

# Forecasting avalanche danger: human-made forecasts vs. fully automated model-driven predictions

Frank Techel[1], Stephanie Mayer[1], Ross S. Purves[3], Günter Schmudlach[2], and Kurt Winkler[1]

[1]WSL Institute for Snow and Avalanche Research SLF, Davos, Switzerland
[2]Skitourenguru GmbH, Zurich, Switzerland
[3]Department of Geography, University of Zurich, Zurich, Switzerland

**Correspondence:** Frank Techel (techel@slf.ch)

**Abstract.** In recent years, the integration of physical snowpack models coupled with machine-learning techniques has become more prevalent in public avalanche forecasting. When combined with spatial interpolation methods, these approaches enable fully data- and model-driven predictions of snowpack stability or avalanche danger at any given location. This prompts the question: Are such detailed spatial model predictions sufficiently accurate for use in operational avalanche forecasting? We
evaluated the performance of three spatially-interpolated, model-driven forecasts of snowpack stability and avalanche danger by comparing them with human-generated public avalanche forecasts in Switzerland over two seasons as benchmark. Specifically, we compared the predictive performance of model predictions versus human forecasts using observed avalanche events (natural or human-triggered) and non-events. To do so, we calculated event ratios as proxies for the probability of avalanche release due to natural causes or due to human load, given either interpolated model output or the human-generated avalanche forecast. Our
findings revealed that the event ratio increased strongly with rising predicted probability of avalanche occurrence, decreasing snowpack stability, or increasing avalanche danger. Notably, model predictions and human forecasts showed similar predictive performance. In summary, our results indicate that the investigated models captured regional patterns of snowpack stability or avalanche danger as effectively as human forecasts, though we did not investigate forecast quality for specific events. We conclude that these model chains are ready for systematic integration in the forecasting process. Further research is needed to
explore how this can be effectively achieved and how to communicate model-generated forecasts to forecast users.

## 1 Introduction

Public avalanche forecasts aim to inform and warn recreational and professional forecast users about the danger of snow avalanches at a regional scale. In many countries, the expected probability of avalanche release, given a specific triggering level, and the potential size of avalanches is described by summarizing this information in one of five avalanche-danger levels
(lowest: 1 (low) to highest: 5 (very high), EAWS, 2023; avalanche.org, 2024). Avalanche danger is then communicated using a mix of formats, including tabular, graphical or text including a mix of symbols, classes, or words (e.g., Hutter et al., 2021). These forecasts are produced by professional forecasters making judgments based on a variety of data sources, including measurements, observations, numerical weather prediction models, and – increasingly – predictions from physically-based snowpack models as *Crocus* or *SNOWPACK* (i.e., Morin et al., 2019). The latter are now often being used in combination with



statistical models or machine-learning approaches (Pérez-Guillén et al., 2022; Fromm and Schönberger, 2022; Hendrick et al., 2023), which aim at making the complex, multi-layered snow-cover simulations more accessible to forecasters by extracting and summarizing information relevant to the forecasting task (e.g., Horton et al., 2019; Herla et al., 2022; Maissen et al., 2024). While forecasting chains have been used for many years, as for instance *SAFRAN-Crocus-MEPRA* in France (Durand et al., 1999), it is now possible to run simulations at much higher spatial and temporal resolutions than those at which forecasters

typically operate by coupling numerical weather predictions models with physically-based snow cover models. These high resolution predictions can therefore also serve as valuable hypothesis testing tools for forecasters in exploring snow-cover conditions and evolution of stability. Moreover, spatially interpolating point or gridded predictions allows predictions for arbitrary points in space and time as well as backcasting for avalanche events at specific locations. In addition to providing reproducible forecasts at higher resolution, model-based forecasting is likely to free up expert time for other tasks – for example,

communicating to professional and recreational mountain users with diverse backgrounds and skills.

To date, distributed snow-cover simulations or interpolated model predictions have been validated using forecaster's best judgments (e.g., in Canada, Herla et al., 2023, 2024) or so-called face validity (Rykiel, 1996), or actual forecasts (e.g., Maissen et al., 2024). Mismatches between scales should be considered when comparing local snow-cover simulations and regional forecasts/judgments. Thus, when model predictions and human judgments/forecasts differ, it often remains unclear whether

forecasters or models were wrong (e.g., Herla et al., 2024). Nonetheless, given numerous recent advances in snow-cover and snow-stability modelling, driven by developments in both physically-based modelling and machine learning, we ask the question: How close is public avalanche forecasting to transitioning from human-driven analysis to fully automated, model-driven methods? This raises the question: Are high-resolution model predictions "good enough" to complement or even replace those made by professional forecasters? To answer this, we need a benchmark that defines what "good enough" means. Given

the challenges in validating avalanche forecasts in general, we define this benchmark through the use of traditional, primarily human-made public avalanche forecasts. Thus, we deem model-driven forecasts to be adequate when they independently make similar forecasts of avalanche danger as expert forecasters.

Public avalanche danger scales are based on the notion that the likelihood, number and size of avalanches increases non-linearly with increasing avalanche danger (levels) (e.g., Techel et al., 2020, 2022; Mayer et al., 2023; Herla et al., 2024). In line

with this fundamental concept of public avalanche forecasting, we evaluate spatially interpolated model predictions and human forecasts using events (avalanches) and non-events, or proxies for non-events focusing primarily on the likelihood of avalanche release. This approach allows us to compare models and human forecasts on objective data. We therefore aim at answering two questions: (1) Is the expected increase in the number of natural avalanches or in locations susceptible to human-triggering of avalanches predicted by spatially interpolated model predictions? and (2) Do fully data- and model-driven predictions achieve

performances comparable to human-made avalanche forecasts?



## 2 Models in support of avalanche forecasting

### 2.1 Recent developments

Recent years have seen rapid growth in use of models aiming to support avalanche forecasting. Based on physical snow-cover simulations using the *SNOWPACK* or *CROCUS* models (Lehning et al., 2002; Vionnet et al., 2012), numerous statistical and

machine-learning models have been developed to provide predictions of potential snow-cover instability (e.g., Mayer et al., 2022), the likelihood of natural avalanche occurrence (Viallon-Galinier et al., 2022; Hendrick et al., 2023; Mayer et al., 2023), the presence and characterization of specific avalanche problems (e.g., Reuter et al., 2022; Herla et al., 2023, 2024), predictions of danger levels (Fromm and Schönberger, 2022; Pérez-Guillén et al., 2022; Maissen et al., 2024) or similarity assessments of simulated snow-cover profiles (Bouchayer, 2017; Herla et al., 2021) allowing spatial clustering of distributed snow-cover

simulations (e.g. Horton et al., 2024). Often these models were trained and validated using observations or judgments made by observers or professional forecasters (e.g., Pérez-Guillén et al., 2022; Herla et al., 2024; Pérez-Guillén et al., 2024). With the aim to support forecasters in their decision-making process, some of these models have been included in operational forecasting processes - for instance in Canada (Horton et al., 2023), France (Morin et al., 2019), or Switzerland (van Herwijnen et al., 2023).

### 2.2 Models used in Switzerland

In the following, we briefly introduce three models, which are used in this study. These models provided live predictions during two forecasting seasons (2022/2023 and 2023/2024) in Switzerland. Avalanche forecasters at the WSL Institute for Snow and Avalanche Research SLF, responsible for producing the national public avalanche forecast, had access to these model predictions during forecast production.

#### 2.2.1 Danger-level model

The *danger-level model*, a random-forest classifier (Breiman, 2001), was trained with a large data set of quality-checked danger levels (Pérez-Guillén et al., 2022). The model uses features describing both meteorological conditions and snow-cover properties simulated with the SNOWPACK model. The classifier predicts the probabilities ($Pr$) for four of the five avalanche danger levels (1 (low) to 4 (high)). This model was live-tested by forecasters during the winter seasons 2020/2021 - 2023/2024.

#### 2.2.2 Instability model

The *instability model* assesses snow-cover simulations provided by the SNOWPACK model with regard to potential instability related to human-triggering of avalanches (Mayer et al., 2022). The random-forest model uses six variables describing the potential weak layer and the overlying slab to predict the probability that a snow layer is potentially unstable. The output probability ranges from 0 (a layer was classified as stable by all the trees) to 1 (classified as unstable by all trees). All simulated layers are assessed using this procedure. In the setup used for forecasting, the layer with the highest probability of instability

($Pr_{\text{instab}}$) is determined and considered as decisive in characterizing this profile, as suggested by Mayer et al. (2022).





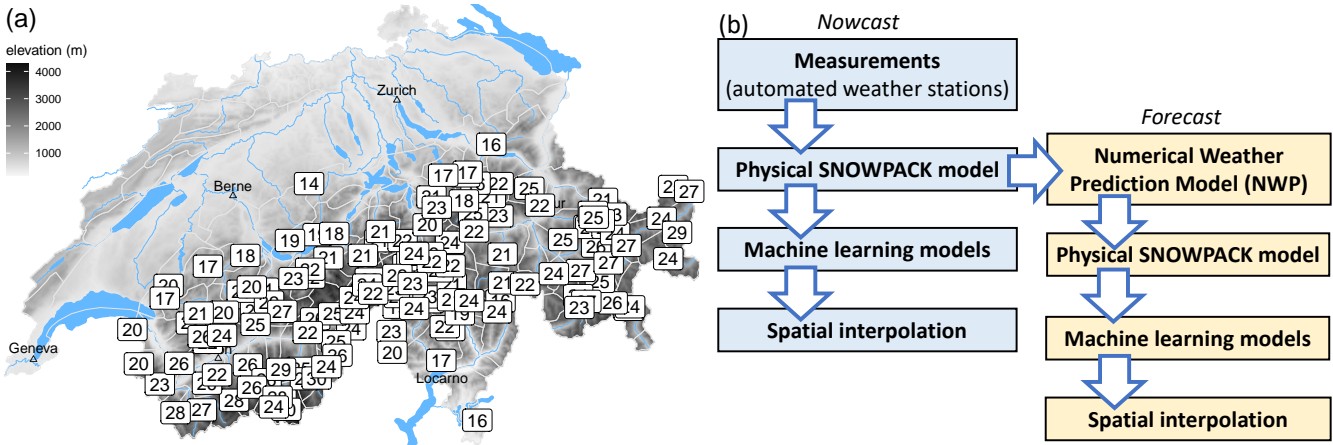

**Figure 1.** (a) Distribution of automated weather stations (AWS) in the Swiss Alps, at which *SNOWPACK* simulations were run. The numbers show the elevation of the station in m a.s.l. divided by 100. (b) Schematic representation of the operational model pipeline for computing the *nowcast* and *forecast* predictions.

### 2.2.3 Natural-avalanche model

The *natural-avalanche model* is a simple one-parameter logistic regression model and comes in several variations: Its input either consists of the 1-day or 3-day sum of new snow or the output of the *instability model* ($Pr_{\mathrm{instab}}$) (Mayer et al., 2023). Trained with a data set of natural avalanches in the vicinity of an automatic weather station (AWS), the models predict the probability of dry-snow avalanches occurring in the same aspect and elevation as the snow-cover simulation. In the operational setup, a weighted mean of the predictions using new snow amounts or $Pr_{instab}$ as input is used by weighting the predictions from the 1-day and 3-day new-snow models with 0.25 and the instability model with 0.5 (Trachsel et al., 2024). We refer to the predicted probability from this weighted approach as $Pr_{\mathrm{natAval}}$.

### 2.2.4 Operational setup in Switzerland

In Switzerland, operationally-used snow-cover simulations are available at the locations of 147 AWS, of which 142 are located throughout the Alps (Fig. 1a). Most of these stations are located at the elevation of potential avalanche starting zones. For *nowcast* predictions, SNOWPACK is driven using hourly measurements obtained from the network of AWS (Fig. 1b). The snow cover is simulated at 3-hour intervals at the location of the AWS for flat terrain and for four virtual slopes (North, East, South, West) with slope angles of 38°, corresponding to typical avalanche terrain. In *forecast* mode (Fig. 1), snow cover simulations are initialized using the most recent *nowcast* simulations (step 2). Simulations are then driven using the COSMO-1 numerical weather prediction model (NWP) with 1 km resolution as input (COSMO = Consortium for Small-scale Modeling (website)), downscaled to the location of the AWS (Mott et al., 2023). This provides forecast snow cover up to 27 hours ahead with a temporal resolution of three hours. ML models provide predictions for flat terrain and for the virtual slopes at the





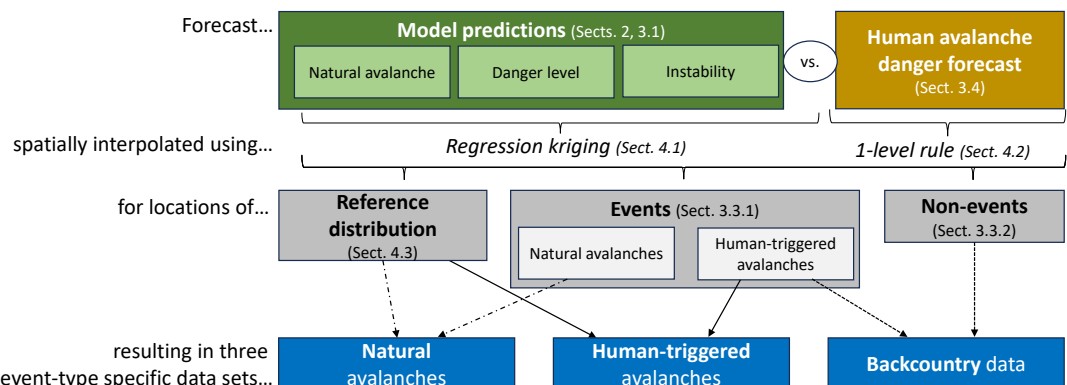

**Figure 2.** Study layout: data and data preparation.

location of the AWS for each of the 3-hour *forecast* and *nowcast* time steps. For interpretation purposes, model predictions are
primarily visualized on maps, sometimes as time series of predictions aggregated by region or elevation. To ease recognition
of spatial patterns, predictions are interpolated in two-dimensional space.

## 3   Data

We used data from the two avalanche forecasting seasons 2022/2023 and 2023/2024, described in detail the following Sections.
Figure 2 provides an overview of the data and methods used for obtaining the three event-type specific data sets.

### 110   3.1   Model predictions

We analyse the model output as described in Section 2.2.4. In all cases, predictions for specific aspects (e.g., N, E, S & W)
are used: the probabilities obtained with the *instability model* ($Pr_{\mathrm{instab}}$) and the *natural-avalanche model* ($Pr_{\mathrm{natAval}}$), and
the predicted probability for danger level $D \geq 3$ (considerable) (*danger-level model*), referred to as $Pr_{\mathrm{D} \geq 3}$. In the latter case,
we opted for $Pr(D \geq 3)$ rather than the predicted danger level $D$ or the continuous expected danger value, as this permitted
analyzing the model in a similar way to the other models though at the cost of loosing some discrimination power at avalanche
conditions representing 1 (low) and (2 moderate) danger. However, $Pr(D \geq 3)$ was strongly correlated to the expected danger
value (see also Appendix Figure A1).

For the purpose of this analysis, we relied exclusively on model predictions calculated in real time during the forecasting
season. Crucially, this means our evaluation is not based on reanalysis data, but rather forecasting of events in an operational
context. For the *forecast* predictions, we used simulations available at 15.00 local time (LT), the time when forecasters meet to
discuss and produce the forecast for the following day. From these, we extracted the prediction valid for the following day at
12.00 LT. In addition to forecast predictions, we also used *nowcast* predictions, allowing us to estimate the effect of biases in
the weather forecast input. For *nowcast* predictions, we extracted the same 12.00 LT time step as for *forecast* predictions. Note



that sometimes data were missing, either because the model was not available at the time (i.e., no data for natural-avalanche
model in *forecast*-mode in 2022/2023 season, as it was only developed in 2023; Mayer et al., 2023), or due to a re-engineering
of the data-model pipeline (no *forecast* predictions for danger-level model for parts of the 2023/2024 season).

## 3.2 Snow-line estimates

The AWS used for avalanche forecasting in Switzerland are primarily located at elevations at or above tree line, with few
situated below 1700 m. Due to the sparsity of data points at elevations below 1700 m, we required an estimation of the
elevation below which there was no continuous snow cover on steep slopes and where therefore no avalanche releases were
possible. To obtain this threshold, we used daily estimates of the approximate elevation above which a continuous snow cover
exists for steep North and South facing aspects as reported by study plot and field observers in Switzerland. This snow line is
reported in 200 m increments. In case the snow line cannot be seen by the observer, no estimate is being made. In total, about
19000 such estimates were available for the two seasons (North and South combined: about 100 per day).

## 135  3.3 Events, non-events and reference distributions

We consider the reported occurrence of an avalanche triggered by natural causes or by human load as an event. Events are
described in detail in Section 3.3.1.

Defining non-events, on the other hand, is much more challenging as non-events are often not reported, and as the absence
of an observed avalanche does not mean there was no avalanche (Hendrick et al., 2023; Mayer et al., 2023). We therefore
followed two paths: (1) we generated reference distributions – described in Section 4.3, representing the range of conditions
over the study period as reference, and (2) we used GPS points as proxies for non-events – described in Section 3.3.2. The
latter can be considered to represent non-events – a person was at a specific point and triggered with rather high certainty no
avalanche. These data have been repeatedly used for this purpose (Sykes et al., 2020; Winkler et al., 2021; Hendrikx et al.,
2022; Degraeuwe et al., 2024). In contrast, using reference distributions limits comparisons to evaluating event conditions
against the full spectrum of possible conditions. Therefore, reference distributions may be particularly suitable to evaluate
the prediction performance for natural avalanches, as these do not require humans to be in avalanche terrain. In contrast, for
human-triggered avalanches, where the presence of humans is required in avalanche terrain, the GPS tracks add another layer
of information. However, these data not only provide information on the presence of humans, they also reflect adjustments in
human behaviour (i.e., choice of ski tour and or slopes skied) due to forecast or encountered avalanche conditions.

## 150  3.3.1 Events: avalanches

In Switzerland, approximately 80 observers or members of local avalanche commissions provide daily reports of avalanches
occurring in their area of observation. Apart from avalanches documented by these observers, additional reports may come from
field observers, who are also part of the observer network, or from the general public. The reported details of avalanches include
their location and estimated time of occurrence, size categorized on a scale of 1 to 5 as per EAWS (2019), moisture content





(classified as dry or wet), and the triggering mechanism (such as natural release or human-triggered), following guidelines
from SLF (2020). Location information generally refers to the top of the starting zone (coordinates, slope aspect, elevation).
For the purpose of this analysis, we consider an **event** to have occurred at the location and date as reported.

**Natural avalanches.** We extracted all avalanches of *size 2 or larger*, classified as a *dry slab* avalanche with trigger type
*natural release*. In total, 1855 avalanches fulfilled these criteria during these two seasons. These were located at a median

elevation of 2505 m (IQR: 2280 - 2676 m).

**Human-triggered avalanches.** For human-triggered avalanches, we considered reported dry-snow avalanches with trigger
type *human*, if the avalanche was either classified as *size 2 or larger* or if a person was caught in the avalanche. As a large share
of these avalanches was reported by the public, we checked the location, size and moisture content for plausibility whenever
possible. In total, during the two seasons, 801 avalanches fulfilled these criteria. Of these, 34% (273) were avalanches with

at least one person being caught. Human-triggered avalanches were located at a median elevation of 2481 m (IQR: 2241 -
2713 m).

### 3.3.2  Non-events: backcountry touring activity (GPS points)

We used a data set of GPS tracks collected on *www.skitourenguru.ch* (website), where users can upload GPS tracks and have
them rated with regard to avalanche risk (Schmudlach and Eisenhut, 2024). 928 different tracks, including time stamps, were

uploaded during the winter of 2023/2024. Since we consider it unlikely that tracks were uploaded if people were involved in
avalanches, we treat these tracks as proxies for non-events. Following post-processing of the GPS tracks – described in detail in
Winkler et al. (2021) and Degraeuwe et al. (2024), this data set contains in total $> 850000$ points. Following largely the criteria
used by Degraeuwe et al. (2024), we extracted points if they were at a distance from controlled ski runs of $\geq 200$ m, if they
were at an elevation $\geq 1600$ m and in potential avalanche terrain, defined by the maximum slope angle within 70 m distance

(for details: Schmudlach, 2022, p. 10) being $\geq 30°$. Lastly, in order to avoid auto-correlation, consecutive points from the same
track had to be $\geq 200$ m apart. The resulting data set comprised 3998 points in avalanche terrain representing backcountry
touring activities with a median elevation of 2298 m (inter-quartile range IQR: 1977 - 2569 m), a slope angle of the steepest
section within 70 m distance of 35° (32 - 39°). The median horizontal distance between any two points from the same track was
566 m (359 - 1039 m) and the difference in elevation 172 m (106 - 285 m). To have a corresponding data set of human-triggered

avalanches, the data-set of human-triggered avalanches described in the previous Section 3.3.1 were also filtered by distance
to ski runs $\geq 200$ m. The resulting spatial distribution of GPS points (non-events) and corresponding avalanches (events) is
shown in Figure 3a.

### 3.4  Avalanche forecast

We extracted the forecast danger level ($D$) and associated sub-level qualifier ($_s$, combined $D_s$) summarizing the severity of

avalanche conditions related to dry-snow avalanches together with the indicated elevation threshold and aspect range from the
avalanche forecast published by *WSL Institute for Snow and Avalanches SLF* (SLF) at 17.00 local time (LT), and valid until
17.00 LT the following day. For danger level 1, no sub-level is available.


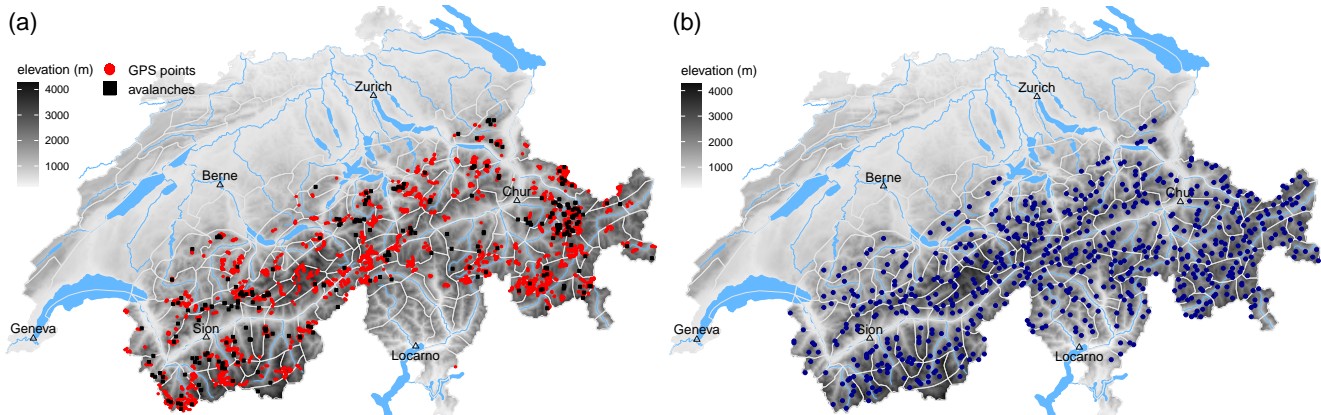

**Figure 3.** Maps of Switzerland showing the spatial distribution of (a) GPS points and human-triggered avalanches (backcountry touring data set) and (b) the randomly sampled subset of grid points used to obtain reference distributions.

The sub-levels have been in use since 2017 (internally) and since Dec 2022 they have been published in the Swiss forecast (Lucas et al., 2023). Using sub-levels allows closer tracking of expected conditions compared to danger levels. On average, a higher forecast sub-level is generally related to more locations susceptible to avalanche release and to more avalanches of larger size (Techel et al., 2022).

## 4 Methods

### 4.1 Spatial interpolation

We spatially interpolated point data (model predictions and snow line estimates) to arbitrary points in avalanche terrain, in our case to the locations of events, non-events and the random subset of grid points used as reference distribution. To do so, we utilized regression kriging[1] (RK), a geo-statistical interpolation method that combines the regression analysis of the dependent variable using additional data and spatial interpolation of the residuals from the regression (Hengl et al., 2007). In our case, we used the location coordinates and the elevation as dependent variables to interpolate model predictions.

Some events or non-events were recorded on North-East, South-East, South-West or North-West aspects. To obtain interpolations for these points, we calculated the respective mean of the $Pr$-values, i.e., for North-East we calculated the mean of the North and East predictions. We proceeded in a similar way with the estimated snow line: for example, the snow line for East aspects was the mean of North and South, the snow line for North-East was a weighted mean of North (weight 0.75) and South (weight 0.25). For locations and elevations, where observers estimated the snow line to be (Sect. 3.2), we set $Pr = 0$.

To find the best-possible kriging settings given the data, we tested several settings using leave-one-out cross validation on a random subset of three cases for each of the three models. Following best practice (e.g., Hengl et al., 2007), we used a

---

[1]see Hengl et al. (2007) for a detailed introduction




logistic transformation of $Pr$ prior to interpolation, setting $Pr = 0$ and $Pr = 1$ to $Pr = 0.001$ and $Pr = 0.999$, respectively. The kriging settings used for interpolation can be found in the $R$-script[2].

## 4.2 Benchmark forecasts: the Swiss avalanche forecast, interpreted using the 1-level rule

We used the forecasts as published in the Swiss avalanche bulletin (Section 3.4) as our benchmark for comparison. To do so, we checked whether a point was within the elevation and aspect range as indicated in the bulletin. If this was the case, we assigned the forecast $D_s$ to this point. If this was not the case, we applied the 1-level rule, subtracting one level from $D_s$ published in the forecast. The 1-level rule is a rule-of-thumb, which has proven reliable to estimate the severity of avalanche conditions outside the indicated aspect and elevation range (SLF, 2023; Winkler et al., 2021). This adjusted danger rating is referred to as $D_s^*$. For the purpose of this analysis, we set $D_s^* = 1$ for cases, when the adjusted $D_s^* < 1$.

## 4.3 Deriving reference distributions for model predictions and avalanche forecast

As we often lacked reliable information for the non-occurrence of avalanches at specific locations (see previous Section 3.3), we created representative reference distributions of $D_s^*$ and model predictions. These reference distributions describe the range of conditions encountered throughout the investigated period without making assumptions about whether an event occurred or not.

To generate these reference distributions, we first defined an artificial data set of points, placed throughout the Swiss Alps, at elevations of interest for avalanche forecasting and winter recreation by randomly sampling 2.5% of the grid points from a 1 km digital elevation model within the elevation range 1600 to 3100 ma.s.l. Within this elevation range, 95% of the natural and human-triggered avalanches were observed, with 1% at lower, and 4% at higher, elevations. Moreover, the critical elevation as indicated in the bulletin essentially always lies between 1800 and 2800 m a.s.l. Applying this filter resulted in 666 points,

with a median elevation of 2203 m a.s.l. (IQR: 1892 - 2507 m a.s.l.). The spatial distribution of these points is shown in Figure 3b. Next, we calculated for each of these grid points and for the four aspects North, East, South, West, the danger level as forecast in the avalanche bulletin applying the 1-level rule (Section 4.2). Similarly to applying the human forecast to these locations, we computed the model-predicted probability using regression kriging (Section 4.1). This was done for all days, for which model predictions were available. Last, we combined model predictions with the avalanche bulletin by location and

date. The resulting data sets contained only cases, when both model predictions and $D_s^*$ were available.

## 4.4 Analysis

We first explored the agreement between the two prediction types (*forecast* and *nowcast*) for each model by calculating the Pearson correlation coefficient ($r$) and the difference between *forecast* and *nowcast* predictions. This allowed us to assess whether systematic bias exist in *forecast* compared to *nowcast* predictions, and to obtain an understanding of the magnitude of

variation between these two.

[2]see link to script repository at end of manuscript





We then addressed our first question, namely whether the models reflect the expected increase in avalanche occurrence probability with increasing model-predicted probability. Binning the model-predicted probabilities ($Pr$) to bins of width 0.05 (for natural and human-triggered avalanches) and of width 0.1 (for backcountry), we counted the number of predictions for representative grid points ($ref$) or GPS track points ($nEv$) and for events ($Ev$) falling into a bin, for each model - prediction-type combination. To investigate whether the probability of avalanche occurrence increases with increasing model-predicted probabilities, we calculated the event ratio $R$ for cases when we relied on the reference distribution:

$$R_{m,i} = \frac{N(Ev)_{m,i}}{N(ref)_{m,i}} \qquad (1)$$

and when using non-events:

$$R_{m,i} = \frac{N(Ev)_{m,i}}{N(Ev)_{m,i} + N(nEv)_{m,i}} \qquad (2)$$

where $N$ is the number of data points in each bin $i$, and for each model and prediction type $m$. To assess whether the increase in $R$ with increasing $Pr$ was monotonic, we calculated the Spearman rank-order correlation ($\rho$) between $R_{m,i}$ and $Pr_{m,i}$.

To address our second objective, the comparison of spatially-interpolated model predictions with our benchmark forecast, the combination of danger level and sub-level ($D_{s}$) interpreted using the 1-level rule ($D_{s}^{*}$, Section 3.4), a further preparatory step was necessary. As the resolution of $D_{s}^{*}$ is limited to a discrete number of classes, we transformed model predictions to reflect these $D_{s}^{*}$-classes by assigning them to bins of size equal to the sub-levels. Though being aware that this may potentially split model predictions in an unfavorable way, we deemed this a generally valid approach, as prior research had shown that model predictions correlated with $D_{s}^{*}$ (Techel et al., 2022). Moreover, this step allowed to directly compare the underlying patterns in $R$ without being distorted by differences in the size of the respective groups.

To obtain bins containing an equal number of data points for human forecasts and for model predictions, we first ordered the model-predicted probabilities from lowest to highest. To assign them to bins in a way that these were of equal size as the corresponding $D_{s}^{*}$-subsets, we derived the respective $Pr$-thresholds for each bin. For example, in the subset containing the predictions for the instability model, the sub-level proportions were $D_{s}^{*} = 1$ (low): 40.9%, $D_{s}^{*} = 2-$: 18.0%, and $D_{s}^{*} = 2 =$: 17.8%. Applying these percentiles to the ordered probabilities of the instability model resulted in thresholds for these three classes of $Pr_{\text{instab}} = [0, 0.266] \rightarrow$ bin 1, $Pr_{\text{instab}} = (0.266, 0.459] \rightarrow$ bin 2, and $Pr_{\text{instab}} = (0.459, 0.710] \rightarrow$ bin 3. Consequently, after splitting the model predictions using these thresholds, the bins contained the same proportion of data points as $D_{s}^{*} = 1$ (low) , $D_{s}^{*} = 2$-, and $D_{s}^{*} = 2$=. For higher sub-levels, we proceeded in the same way. In a second step, applying the same thresholds, we calculated the number $N$ of nEv and Ev falling into each bin. Similar to before, we then calculated the event ratio $R_{m,i}$.

For visualisation purposes, we derived a relative ratio, by normalizing individual $R_{m,i}$-values using the overall base rate event ratio $R_{\text{m}}$, defined as

$$R_m = \frac{N(Ev)_m}{N(ref)_m}, \qquad (3)$$

when relying on the reference distribution, and

$$R_m = \frac{N(Ev)_m}{N(Ev)_m + N(nEv)_m} \qquad (4)$$





for non-events. From these, we calculated the relative ratio as

$$RR_{m,i} = \frac{R_{m,i}}{R_{\mathrm{m}}}. \tag{5}$$

The factor by which $RR$ increases between two consecutive bins (i.e., from bin 1 to bin 2, or from 1 (low) to 2-) describes how well sub-levels ($D_s^*$) and model predictions discriminate between neighbouring bins/sub-levels. We therefore derived the median of the factors $F$ over all consecutive bins summarizing how well sub-levels ($D_s^*$) and model predictions discriminate on average. In addition, we compared $RR$-values for human forecasts and model predictions using a *Chi-Square test*.

## 5 Results

Before addressing the two research questions, we compared *nowcast* and *forecast* predictions and the correlation between different model predictions (detailed results can be found in Table A1 in the Appendix). *Nowcast* and *forecast* predictions of the same model showed very high correlations ($r \geq 0.92$). No clear pattern emerged with regard to a bias. The correlation between different models was generally high ($r \geq 0.68$). The correlation between the natural-avalanche model and the instability model was lowest ($r \leq 0.70$), the correlation between danger-level model and the instability model the highest ($r \geq 0.88$).

### 5.1 Model-predictions

Figure 4 summarizes model predictions by model (danger level, instability, natural avalanche) and prediction type (forecast, nowcast), separated for natural avalanches (left column), human-triggered avalanches (middle column), and data related to back-country touring.

#### 5.1.1 Reference distributions and non-events

Examining the reference distributions shows that low $Pr$-values of the natural-avalanche model were dominant, with almost 60% of the predictions at $Pr_{\mathrm{natAval}} < 0.05$, and increasingly fewer occurrences at higher $Pr$-values (Figure 4a). Similar though less pronounced patterns can be noted for the danger-level model ($Pr_{\mathrm{D}\geq 3} < 0.05 \approx 35\%$) and the instability model ($Pr_{\mathrm{instab}} < 0.05$: $15 - 20\%$), which is best seen in Figure 4b. Deriving the respective median value from the reference distributions shows that on average the probability for natural avalanches is low in the large majority of cases ($Pr_{\mathrm{natAval}} = 0.04$ in *forecast*-mode, see Table 1). The median value for the danger-level model ($Pr_{\mathrm{D}\geq 3} = 0.12$) corresponds to about the threshold between a predicted danger level 1 (low) and 2 (moderate) (see also Figure A1b in Appendix), while for the instability model the median ($Pr_{\mathrm{instab}} = 0.31$) would be classified as *stable* according to the classification proposed by Mayer et al. (2022). Comparing the distribution of non-events (GPS tracks, Figure 4c) with the reference distribution in Figure 4b shows similar patterns and comparable median values (Table 1).

**Figure 4.** Model predictions. Columns show the respective results for (left) natural avalanches, (middle) human-triggered avalanches, and (right) for data stemming from back-country touring activities. Upper row: (a, b) reference distributions, simulated on the representative grid and (c) non-events(GPS points); middle row: events with (d) natural avalanches, (e) human-triggered avalanches and (f) human-triggered avalanches during backcountry touring; lower row: (g-i) event ratios. Shown are the results for each model (colour) and prediction type (shape). To allow better comparison between models, proportions rather than absolute numbers are shown, where 100% relates to the numbers shown in Table 1.





**Table 1.** Data availability and median probabilities ($Pr$) for events ($Ev$), reference distribution ($ref$) or non-events ($nEv$) for the respective models, prediction types and data subsets.

| event type | model | prediction type | days | $N$ $ref$/$nEv$ | $Ev$ | $Pr$ $ref$/$nEv$ | $Ev$ |
|---|---|---|---|---|---|---|---|
| natural avalanches | natural avalanche | nowcast | 283 | 672383 | 1754 | 0.03 | 0.45 |
| | | forecast[a] | 143 | 351964 | 855 | 0.04 | 0.42 |
| | instability | nowcast | 283 | 672383 | 1754 | 0.30 | 0.80 |
| | | forecast | 236 | 562475 | 1554 | 0.31 | 0.80 |
| | danger level | nowcast | 298 | 711023 | 1791 | 0.14 | 0.76 |
| | | forecast[b] | 219 | 516735 | 1435 | 0.12 | 0.74 |
| human-triggered avalanches | instability | nowcast | 283 | 672383 | 737 | 0.30 | 0.74 |
| | | forecast | 236 | 562475 | 648 | 0.31 | 0.76 |
| | danger level | nowcast | 298 | 711023 | 762 | 0.14 | 0.60 |
| | | forecast[b] | 219 | 516735 | 621 | 0.13 | 0.60 |
| backcountry[a] | instability | nowcast | 129 | 3309 | 260 | 0.37 | 0.76 |
| | | forecast | 124 | 3173 | 244 | 0.36 | 0.77 |
| | danger level | nowcast | 129 | 3309 | 260 | 0.15 | 0.59 |
| | | forecast[b] | 73 | 1642 | 176 | 0.14 | 0.58 |

Data availability: [a] - only in 2023/2024, [b] - 2022/2023 and from Feb 2024

## 5.1.2 Events

The distribution of events showed different patterns (Figure 4d-f) between the models but also compared to the reference distributions and non-events (Figure 4a-c).

While natural avalanche events were approximately similarly distributed across the entire range of $Pr_{\mathrm{natAval}}$-values (Figure 4d), the number of natural avalanches increased considerably with increasing $Pr$-values for the other two models. The median value was $Pr_{\mathrm{natAval}} = 0.42$, indicating that on average the model predicted almost a 50% chance of a natural avalanche occurring. The median values for the danger-level model $Pr_{\mathrm{D} \geq 3} = 0.8$ and instability model $Pr_{\mathrm{instab}} = 0.74$ were high, which correspond to a model-predicted danger level well within danger level 3 (considerable) (Figure A1) and to about the threshold between profiles classified as potentially unstable according to the classification by Mayer et al. (2022).

Human-triggered avalanches were more frequent when the models predicted higher probabilities (Figure 4e, f). This pattern was much more pronounced for the instability model, with particularly many events when $Pr_{\mathrm{instab}} \geq 0.9$. The median values for human-triggered avalanches were similar for the subsets containing all human-triggered avalanches (Fig. 4e) and the subset



of events during backcountry touring activities (Fig. 4f). $Pr$-values were lower for human-triggered avalanches compared to natural avalanche events (Table 1).

$Pr$-values differed significantly between events and non-events or reference distributions across all models and data subsets, regardless whether these were calculated in *nowcast-* or *forecast*-mode (Wilcoxon rank-sum test: $p < 0.001$).

### 5.1.3 Event ratio

The ratio $R$ between the number of events in relation to the reference distributions (Eq. 1) or to the number of non-events (Eq. 2), referred to as event ratio, provides an answer to our first research question, as it shows whether model-predicted probabilities capture the expected increasing frequency of (potential) triggering locations. As can be seen in Figure 4g-h, the ratio $R$ increased strongly, and in some cases in an exponential fashion, with increasing $Pr$ for all models and data sub-sets (Spearman rank-order correlation $\rho \geq 0.9$).

Mayer et al. (2022) suggested thresholds to classify predictions by the instability model into predictions indicating stability ($Pr_{\mathrm{instab}} < 0.5$), potential instability ($Pr_{\mathrm{instab}} \geq 0.77$), and potential instability but with a high false-alarm rate ($Pr$-values in between). In the backcountry touring data set, the ratio $R$ for human-triggered avalanches was 5.1 times higher when the instability model (*forecast*-mode) indicated potential instability compared to the model predicting stable conditions, and 2.4 times higher compared to the in-between class. While using these three classes may help in interpreting model outputs, the resulting coarse classification clearly results in a loss of discriminatory power.

### 5.2 Comparison with benchmark forecast, the avalanche bulletin

We now compare the model predictions (*forecast* mode) with the avalanche bulletin. To make this comparison possible, we assigned the rank-ordered model-predicted probabilities to bins containing equal proportions of data points as the sub-level distributions in the bulletin (described in Section 4.4), from which we derived $R$. As there were (almost) no data points for backcountry touring data (human-triggered avalanches, GPS points) at $D_s^* > 3+$, $R$ could not be calculated. In contrast, for natural avalanches a sufficiently large number of data points was available even for the highest bins. As a consequence, the number of bins varies between the three event-type specific data sets, ranging between seven bins (from 1 (low) to $D_s^* \geq 3+$) for the backcountry data set (Fig. 5c) and ten bins (from 1 (low) to $D_s^* = 4+$) for natural avalanches (Fig. 5a). To ease interpretation of the event ratios $R$ and to make them comparable across data sets and models, these were normalized using the base-rate event ratio (Eq. 5). Figure 5 shows the resulting relative ratios $RR$, while additional information on the respective distributions of events and non-events is provided in the Appendix (Figures B1 - B3).

Overall, models and bulletin showed generally monotonically increasing relative ratios. Exceptions were the respective two highest bins for the human forecast for the human-triggered avalanches (Fig. 5b) and for the models in the backcountry data (Fig. 5c). The larger scatter between models in Figure 5c and the drop in $RR$ in the respective highest bin (Fig. 5b, c) is likely due to the combination of few data points causing greater variability and the fact that the data is influenced by human behaviour.





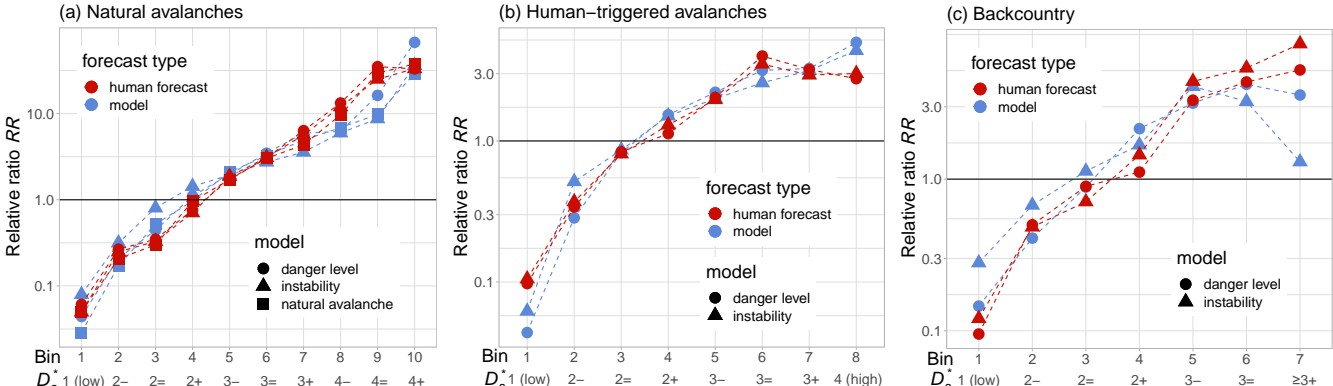

**Figure 5.** Ratio of events (avalanches) to (a, b) reference distribution of model predictions or (c) non-events (GPS tracks), normalized by the overall base-rate ratio of events (Eq. 5) for (a) natural avalanches, (b) human-triggered avalanches, and (c) the backcountry data set. Note the log-scale on the y-axis.

$RR$ increased most strongly for natural avalanches (Fig. 5a), with the (median) factor $F$ describing the increase from one
bin to the next higher one ranging between 1.67 and 2.1, and a total increase between the highest and lowest bins by a factor $F$
between 455 and 1520 (Tab. 2). For human-triggered avalanches and for the backcountry data (Fig. 5b, c), the median increase
between bins ranged between $F = 1.39$ and $F = 1.82$. While the median increase is a robust measure describing the average
$RR$-difference between neighbouring bins and not susceptible to individual extreme values, the factor describing the increase
from lowest to highest bin is highly sensitive to small variations in the respective lowest and highest bins. For example, for the
instability model in the backcountry data set $F$ was merely 5, much lower than all the other values (Tab. 2). The highest bin for
this model contained only two events and 20 non-events. If there would have been just one more event in this bin, this would
result in $F = 7$, five events more would result in $F = 16$. Thus, the factor describing the total increase between the respective
highest and lowest bins are indicative at best. Comparing the corresponding curves between model predictions and human
forecasts showed no significant differences (chi-square-test: $p > 0.05$). In summary, model predictions and human forecasts
exhibited similar levels of discriminatory ability between bins.

## 6  Discussion

We analyzed the performance of three spatially-interpolated models predicting the probability of avalanche occurrence due
to natural causes or due to human load, and showed that increasing model-predicted probabilities correlated positively and
strongly with the event ratio $R$ (Figure 4), which we consider a proxy for the probability of avalanche release. Moreover, we
showed that model and human forecasts, interpreted using the 1-level rule, reach approximately equal performance in terms of
discriminating between sub-levels or sub-level equivalent model-predicted probabilities (Figure 5). However, the underlying
data and applied methods are prone to several limitations, which we discuss first, before reflecting on the interpretation of




**Table 2.** Factor $F$ summarizing the increase between any two neighbouring bins (Figure 5). Shown are the median increase, and the factor between the respective highest bin/sub-level and lowest bin/sub-level.

| event type | model | F (median) bulletin | models | F (highest/lowest) bulletin | models |
|---|---|---|---|---|---|
| natural avalanches | danger level | 2.09 | 2.10 | 540 | 1520 |
| | instability | 2.08 | 1.67 | 666 | 455 |
| | natural avalanche | 1.79 | 1.86 | 748 | 1013 |
| human-triggered avalanches | danger level | 1.80 | 1.53 | 28 | 114 |
| | instability | 1.60 | 1.39 | 29 | 71 |
| backcountry | danger level | 1.56 | 1.82 | 55 | 25 |
| | instability | 1.75 | 1.58 | 65 | 5 |

these findings in light of assumptions we made. Finally, we discuss implications for avalanche forecasting more generally with respect to the adoption of model-driven forecasting processes.

## 6.1 Limitations and assumptions

The data-sets of natural and human-triggered avalanches and GPS tracks represent only a small fraction of actual activity. For example, Degraeuwe et al. (2024) estimated that the GPS tracks in the dataset represent the activity of only about 1 in 2000 backcountry users. Moreover, events are likely not missing-at-random but are related to factors such as visibility, avalanche size, and the severity of incidents (e.g., Jamieson and Jones, 2015; Mayer et al., 2023). Moreover, there is uncertainty related to the exact location and timing of avalanches. For human-triggered avalanches, starting-zone coordinates and release date were checked for plausibility (and corrected if needed) during the seasons, for natural avalanches this was not possible.

We analyzed the predictions obtained from the operational model pipeline in real time. We made no attempts to improve any part of the pipeline or to remove outliers as these errors are part of the pipeline as are human-made errors in the case of the human forecasts.

We focused on the probability of avalanche occurrence, either due to natural causes or related to human-triggering. Avalanche size, which is expected to increase with increasing danger level (Schweizer et al., 2020; Techel et al., 2020), was not analyzed in detail. Avalanche size, however, is reflected in human forecasts and is therefore also implicitly contained in the predictions by the danger-level model, as this model was trained using a historic data set of quality-checked avalanche forecasts (Pérez-Guillén et al., 2022). In contrast, both the natural-avalanche model and the instability model were trained with a focus on estimating the probability of avalanche release due to natural causes or due to a human load.

For the purpose of this analysis, we assumed that the 1-level rule is a good approximation to apply the information provided in the human avalanche forecast to locations outside the aspects and elevations indicated in the public avalanche forecast. Even





though this rule-of-thumb has been used for many years to apply the bulletin to avalanche terrain during the planning phase of ski tours (e.g., SLF, 2023), there are likely more suitable approaches, which reflect the more gradual – rather than step-wise – increase of avalanche danger with elevation and aspect (Winkler et al., 2021; Degraeuwe et al., 2024). At the same time, for the comparison of model predictions with human forecasts, we assigned rank-ordered, model-predicted probabilities to bins equal in size to the proportion of sub-levels. While this facilitated the comparison, it possibly split model predictions in an unfavorable way, potentially reducing discrimination capabilities of model predictions.

For part of this analysis, we relied on data reflecting human behaviour in avalanche terrain, which is known to be impacted by avalanche conditions (e.g., Winkler et al., 2021). If humans were fully ignorant of conditions and did not change their behaviour in response to forecasts or encountered conditions, the event ratio $R$ would represent the probability of avalanche release due to human load. In contrast, if humans were perfectly able to detect all locations susceptible to avalanche release and avoid these, we could not use this data as a proxy for the probability of avalanche release. However, as reality lies somewhere in between these two extremes, we consider $R$ to be a suitable proxy even though we do not know to what degree human behaviour impacts $R$-values, and whether and how this differs between model and human forecasts.

## 6.2 Comparison with similar studies

Previously, several studies have shown that model predictions correlated with human forecasts considering danger levels, sub-levels, but also aspect and elevation information (e.g., Techel et al., 2022; Mayer et al., 2023; Herla et al., 2024; Pérez-Guillén et al., 2024). In all these studies, point predictions were compared with the regional forecast or human judgments. Here, we applied the information provided in the human forecast and the model predictions to the exact location of events or non-events, similar to the studies by Winkler et al. (2021) and Degraeuwe et al. (2024), who analyzed large data sets of backcountry touring activity using the avalanche forecast as input. They observed an increase in the chance to trigger and be caught in an avalanche with increasing danger level, calculated as in Equation 2, resulting in avalanche risk increasing in a similar fashion as our findings. Our results are also in line with Soland (in prep.), who explored spatial predictions of the instability model in *nowcast*-mode using a multi-year data set of GPS tracks and human-triggered avalanches. For instance, Soland obtained similar median values for the instability model for non-events (2 years: $Pr_\text{instab} \approx 0.35$) and for events (4 years: $Pr_\text{instab} \approx 0.75$; compare to Table 1). Similarly, in a pilot study comparing the predictions of the natural-avalanche model with avalanches detected using automated detection systems for two systems in Southwest Valais (Switzerland), Trachsel et al. (2024) obtained median $Pr_\text{natAval} = 0.50$ for events and $Pr_\text{natAval} = 0.13$ for non-events. Again, these values are similar to what we observed.

## 6.3 Verification of distributed predictions of rare and severe events

Avalanches are generally rare but potentially severe events. Exceptions are situations of widespread instability or when avalanches are very small. Public avalanche forecasts communicate the probability of these rare and severe events in a region through danger levels, or by using symbols or narrative text descriptions (e.g., EAWS, 2023; Hutter et al., 2021). They can therefore be considered a type of *rare and severe event forecast* (RSE), following the notion of Murphy (1991). Verifying RSE forecasts is particularly challenging due to the rarity of events, their localized nature, and the mismatch in scales between regional forecasts





and local events. In practice, for a specific point in avalanche terrain within a region, the probability of avalanche occurrence is very low in most cases. We accommodated these challenges by interpolating to specific points and by evaluating the discriminatory power of human forecast and model predictions considering the increase in event ratio with increasing sub-level or model-predicted probability rather than by classifying forecasts and predictions using absolute terms as 'right' or 'wrong'.

By doing so, we avoided comparing (distributed) model predictions relying on forecasters' best judgments as ground truth as is often done due to a lack of objective data (e.g., Herla et al., 2024).

Avalanche records are indicators of events; unfortunately, these are notoriously incomplete (e.g., Hafner et al., 2021). Automated avalanche detection systems using ground-based or airborne technologies have the potential to allow a much more systematic and continuous detection of events (e.g., Eckerstorfer et al., 2016; Fox et al., 2024; Hafner et al., 2022), particularly

with regard to occurrence and absence of natural avalanches. However, the avalanche detection rate is impacted by avalanche properties including the type (wet or dry) and size of avalanches (e.g., Mayer et al., 2020; Hafner et al., 2021). Nonetheless, these systems likely provide the best means for obtaining increasingly complete avalanche records in the future, though they still do not resolve the issue of recording non-events under additional loads.

While an avalanche is a clear and objective indication that the snowpack was susceptible to triggering given a certain

triggering mechanism (i.e., natural causes or additional loads from human activities) at the location and time of release, it is conceptually more challenging to be certain of non-events, as these require continuous monitoring of avalanche activity at a specific location, and - in case of triggering given additional loads such as a skier - this also requires knowledge about whether a person skied a slope without releasing an avalanche. To our knowledge, GPS tracks are currently the most-widely used means to track actual terrain choices of recreationists (e.g., Sykes et al., 2020; Winkler et al., 2021; Degraeuwe et al., 2024), and

notionally also provide information on non-events. Note though that near misses or sloughs may have occurred, so these data are at best a proxy.

### 6.4 Spatially highly-resolved predictions

As SNOWPACK simulations are driven at the location of automated weather stations (AWS) in Switzerland, we generated spatially distributed predictions by interpolating these point predictions. Although we interpolated to very specific locations

– such as the exact coordinates of the avalanche start zones, it is important to emphasize that interpolation can only provide regional patterns of individual parameters and not for specific slopes. As can be seen in Figure 6, keeping elevation and aspect fixed, interpolation results in primarily larger-scale patterns. Differences within a region are either related to differences in the elevation of grid points – the elevation gradient is modelled as part of regression kriging, and slope aspect – representing variations in aspect-specific snow-cover simulations.

In warning services in other countries, SNOWPACK simulations are directly driven on NWP grids for avalanche forecasting purposes (e.g., in Canada, Herla et al., 2024). Running grid-based snow-cover simulations directly at many locations offers the opportunity to analyze all simulated features at many more points, allowing for instance characterizing the type, stability and depth of simulated weak layers in a region or for specific grid-points (e.g., Herla et al., 2024). While this is an advantage compared to simulating the snow cover at a small number of AWS, there is a clear benefit of forcing SNOWPACK simulations





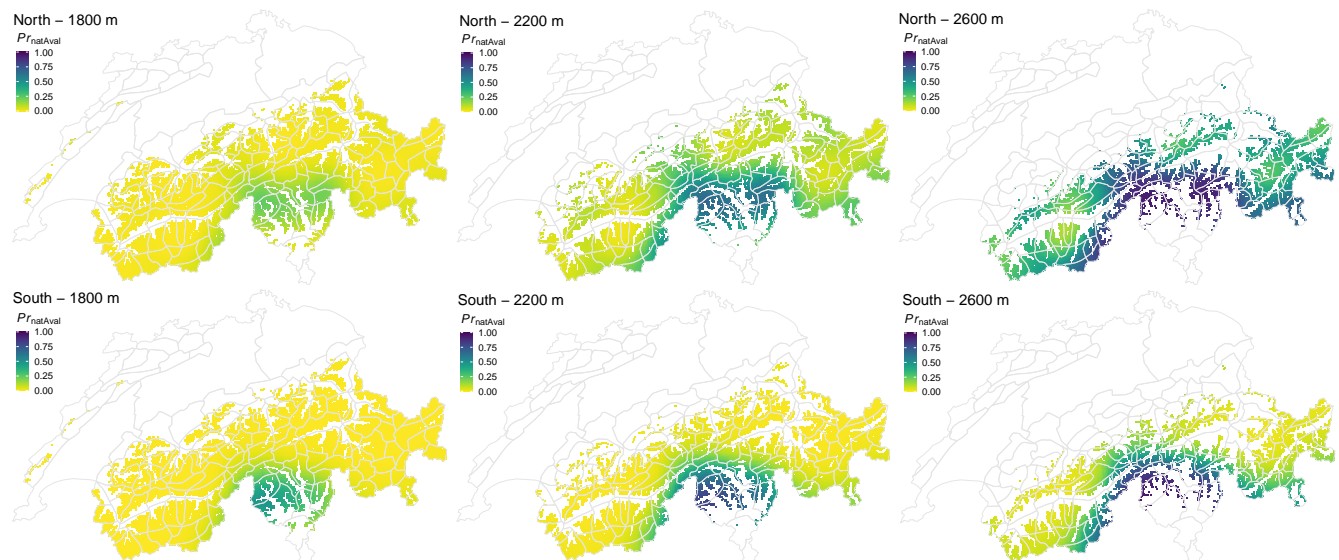

**Figure 6.** Interpolated predictions (natural-avalanche model, forecast-mode), valid for 11 March 2024 for three elevations (left: 1800 m, middle: 2200 m; right: 2600 m) and two aspects (upper row: North, lower row: South). For visibility, grid points 400 m below and all points above the indicated elevation are coloured. The elevation of these grid points is held fixed.

with data from AWS up to the actual time and then adding the NWP forecast data to produce a forecast (setup described in Sect. 2.2.4) as the resulting snow cover simulations have greater similarity to reference profiles than simulations driven exclusively with NWP data (Herla et al., 2021; Binder et al., 2024). Similar observations were made when comparing predictions and actual occurrences of wet-snow avalanche activity. Again, the currently used forecasting-approach in Switzerland showed better correlations than a purely NWP-driven setup (e.g., Bellaire et al., 2017).

## 6.5 Human vs machine, or: human and machine?

We deem model-driven forecasts to be adequate when they independently make similar forecasts of avalanche hazard to those produced by an expert team. Keeping in mind the limitations related to data and methodology, our results suggest that human-made forecasts and model predictions discriminate similarly (well) between conditions considered to be generally stable (i.e., $D = 1$ (low) or bin 1) and those considered the most susceptible to avalanche release. Note, however, that forecasters had access

to model predictions during forecast production and we assume that some of the information provided by the models already impacted the avalanche forecast. In contrast, no such information leakage existed the other way round. This means that we compared purely data-driven, spatially-interpolated *model predictions* to *human-made forecasts including model predictions* interpreted using the 1-level rule. Moreover, while models only used meteorological measurements to correct for potential *forecast* errors, forecasters integrated avalanche observations and other field observations to assess current avalanche conditions. In

summary, currently, the team of two or three human forecasters utilizing all available data and jointly producing the avalanche





bulletin at SLF seems to perform about as well as a model pipeline with no access to additional verification data. However, in this study we did not investigate specific situations, which may be missed by models due to a lack of similar conditions represented in the training data. These situations may be important but infrequent, and will therefore hardly influence the global performance measures. Thus, we see a need for further studies utilizing different data sets, ideally from other forecasting areas,

and methods, and focusing on rare conditions. Furthermore, future models designed to predict snow instability or avalanche danger should incorporate features that reflect the latest advances in understanding the physical processes behind avalanche formation. By integrating more physics into these models, predictive performance can be improved, even in conditions not represented in the training data.

## 6.6 Outlook and future directions

Our results show that the performance of avalanche forecasting model chains has increased considerably in recent years, reaching a level where these models can achieve a performance comparable to that of human-made regional avalanche forecasts, when interpreted using a simple 1-level rule.

It is evident from this study, as well as from several other recent studies (e.g., Herla et al., 2023, 2024; Techel et al., 2022; Pérez-Guillén et al., 2024; Maissen et al., 2024; Trachsel et al., 2024), that the time has come to integrate forecasting models

more systematically into the avalanche forecasting process. We therefore suggest that fully data- and model-driven forecasting pipelines become an integral part of avalanche forecasting. The integration of model predictions may be done either by relying on models as an additional data-source, by utilizing models to summarize relevant information (e.g., Horton et al., 2024), or by providing independent "second opinions" valuable for the decision-making process (e.g., Purves et al., 2003; Maissen et al., 2024; Winkler et al., 2024). In the future, as model performance continues to improve and eventually surpasses that of human

forecasters, the shift to increasingly automated avalanche forecasting may become a possibility. To ensure that predictions are closely aligned with actual conditions, additional data sources must be integrated into model prediction pipelines - as for example, information from real-time avalanche detection systems. Moreover, it must be ascertained that models are capable of predicting out-of-the-box situations, for which they had no training.

Given the rapid growth of models to support avalanche forecasting, we believe that avalanche forecasts can be produced at

greater spatial and temporal resolutions in the coming years. However, the resolution of such forecasts must correspond to the resolution that can be reasonably achieved given the available data. Moreover, spatial clustering of highly-resolved predictions will be needed to effectively and efficiently communicate avalanche conditions to forecast users. More generally, forecasters may spend more time explaining and communicating forecast outputs than generating them in the future.

Increasingly higher-resolved, automated predictions will eventually also become available to users. In theory, such informa-

tion could already be provided to users, and – considering this study, with little reduction in prediction performance compared to human forecasts. However, how will forecast users interpret such spatially highly-resolved information, which is still regional and not slope-specific? The platform skitourenguru (website) already provides one potential route to communicating location-specific risk ratings of avalanche hazard through coupling the human forecast (regional scale) with highly-resolved terrain information (Winkler et al., 2021; Degraeuwe et al., 2024).





Our research suggests that the data, computing power and modelling techniques have finally reached a point where avalanche forecasts can be produced which are broadly comparable with those created by humans. These developments mimic those in many other fields of human endeavour generally, and most specifically with respect to weather forecasting, where operational forecasting production has undergone a revolution in recent years (Young and Grahame, 2024). It will be important to discuss the limitations of using machine-learning approaches in avalanche forecasting (e.g., in forecasting rare events with very limited

training data and modelling conditions not found in training data), developing methods to communicate higher resolution information, and defining the future role of human forecasters in avalanche forecasting. Furthermore, resilience of forecasting chains with respect to the potential unavailability of computational resources or automated measurements requires that human forecasting skill is maintained and developed.

## 7 Conclusions

We have shown that three spatially-interpolated models predicting avalanche danger, the probability of avalanche release, and snowpack instability are capable of predicting expected increasing probabilities of avalanche release due to natural causes or human load. Moreover, the predictive performance of these model predictions are broadly comparable to the performance of human forecasters in an operational setting. Thus, fully data- and model-driven avalanche forecast pipelines – such as the ones discussed in this study, are ready to become an integral part of the avalanche forecasting process, mimicking changes to

operational weather forecasting which have occurred over the last decades. Based on these findings, we conclude that public avalanche forecasting may be reaching a point where a transition from primarily human-made forecasts to machine-generated forecasts is appropriate. The extensive network of real-time data and observations in Switzerland, coupled with high resolution weather forecasting model output, may provide a particularly appropriate setting for such developments. Nonetheless, more work is needed, including improving each step of forecasting pipelines, reliably predicting infrequently-occurring conditions,

validating distributed or spatially-interpolated predictions, optimally integrating models in the forecasting process, and – lastly, but crucially – effectively communicating spatially and temporally highly-resolved forecasts and their uncertainties to forecast users.

*Code and data availability.* The data is available at the data repository: 10.16904/envidat.535. The *R*-Code will be made available at: https://gitlabext.wsl.ch/TechelF/model-driven-avalanche-forecasting-evaluation-study-nr1.



# Appendix A: Appendix

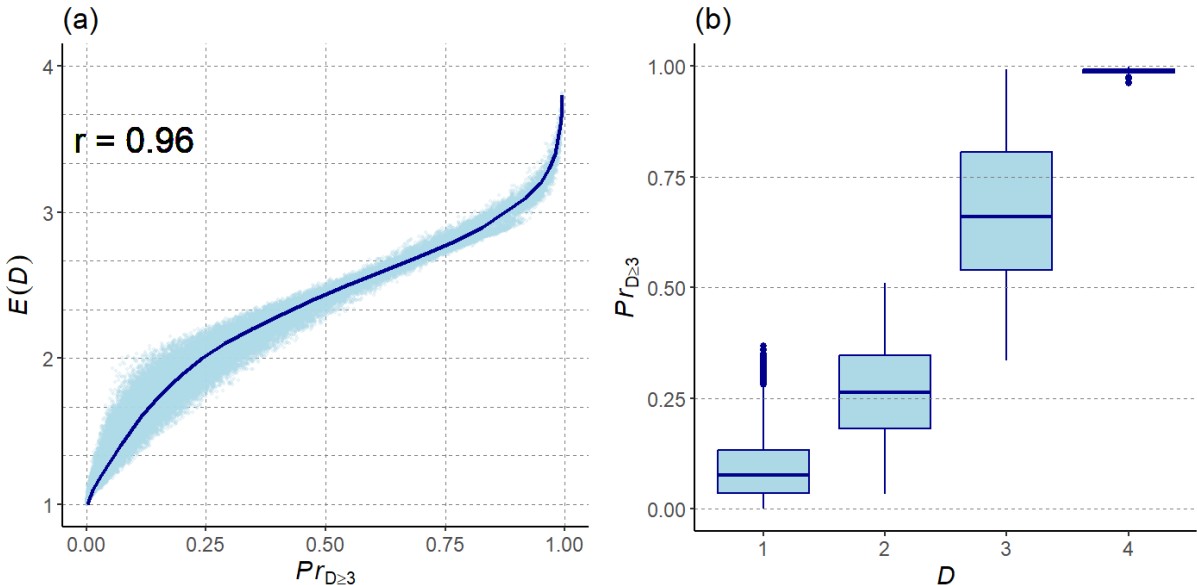

**Figure A1.** Relationship between (a) $Pr_{D \geq 3}$ and the expected danger value $E(D)$ and (b) the model-predicted danger level $D$ and $Pr_{D \geq 3}$ for the 56000 predictions of the danger-level model in *forecast*-mode during the 2022/2023 season.

**Table A1.** Spearman correlation $r$ between models and prediction types (*forecast*, *nowcast*). For the comparison between prediction types for the same model, the mean $\mu$ and standard deviation $sd$ are shown.

| model | prediction type | correlation | difference |
|---|---|---|---|
| danger | forecast-nowcast | $r = 0.98$ | $\mu = 0.005, sd = 0.06$ |
| instab | forecast-nowcast | $r = 0.97$ | $\mu = 0.0004, sd = 0.08$ |
| natAval | forecast-nowcast | $r = 0.92$ | $\mu = -0.001, sd = 0.09$ |
| danger-instab | forecast | $r = 0.90$ | |
| danger-natAval | forecast | $r = 0.92$ | |
| instab-natAval | forecast | $r = 0.71$ | |
| danger-instab | nowcast | $r = 0.88$ | |
| danger-natAval | nowcast | $r = 0.85$ | |
| instab-natAval | nowcast | $r = 0.68$ | |

*Author contributions.* FT - Conceptualization, Methodology, Data Curation, Formal Analysis, Writing - Original Draft; SM - Methodology, Writing - Review and Editing; RP - Writing - Review and Editing; GS - Data Curation, Writing - Review and Editing; KW - Methodology, Writing - Review and Editing



**Figure B1.** Natural avalanches. Left column: model predictions, right column: human avalanche forecast. Upper row: reference distributions, middle row: events (natural avalanches), bottom row: relative ratio $RR$ (Eq. 5).




**Figure B2.** Human-triggered avalanches. Left column: model predictions, right column: human avalanche forecast. Upper row: reference distributions, middle row: events (human-triggered avalanches), bottom row: relative ratio $RR$ (Eq. 5).

**Figure B3.** Backcountry touring data. Left column: model predictions, right column: human avalanche forecast. Upper row: non-events (GPS tracks), middle row: events (human-triggered avalanches, subset backcountry touring), bottom row: relative ratio $RR$ (Eq. 5).





*Competing interests.* No competing interests.

*Acknowledgements.* We thank Marc Ruesch and Andrea Helfenstein, who implemented the operational SNOWPACK - model pipeline during 2023/2024, including the three models described in Section 2. We also benefited from Katia Soland working in parallel on her Master thesis (supervised by FT and RP), in which she explored kriging algorithms using the instability model and a larger data set of backcountry touring data (Soland, in prep.). Marc Ruesch, Andrea Helfenstein, Cristina Pérez-Guillén, and Katia Soland provided feedback on an extended abstract of this manuscript, submitted to the *International Snow Science Workshop 2024* in Tromsø, Norway (Techel et al., 2024).



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
