# Peer review of "Can model-based avalanche forecasts match the discriminatory skill of human danger level forecasts? A comparison from Switzerland"

_Natural Hazards and Earth System Sciences, 2024_

## Referee Comment (RC2)

[referee-annotated manuscript omitted]

---

## Author Comment (AC1)

**Response to review by Florian Herla**

Frank Techel, and co-authors

Dear Dr. Florian Herla,

we greatly appreciate your detailed and constructive feedback to our manuscript. Please find our responses below (in blue).

**1  Summary**

The manuscript titled "Forecasting avalanche danger: human-made forecasts vs. fully automated model-driven predictions" presents a novel approach for evaluating the performance of avalanche hazard assessment models, a highly relevant topic within the scope of NHESS. By leveraging data sets of natural and human-triggered avalanches as well as GPS tracks of backcountry users, the authors pursue a statistical exercise to address two main question. (1) Can spatially interpolated model predictions of avalanche danger and snowpack stability reflect observed avalanche activity, and (2) How well do the automated predictions perform relative to human-made avalanche bulletins. The authors conclude that the model-predicted probabilities correlated strongly with their proxy variable for the probability of avalanche release, and that the model predictions discriminate between different avalanche hazard situations as well as the human-made bulletins. These are substantial findings that the authors introduce and discuss well in the context of underlying assumptions, existing literature, and the future of avalanche forecasting.

I have one main comment that could help make the manuscript even stronger. The comparison between human and model performance in discriminating between different conditions (Fig. 5) is not completely independent. In L254ff the authors explain how the model predictions are tied to the human predictions, and in L381–383 they discuss that this could reduce the estimated model performance. To make the present approach more transferable to other countries and the results more illustrative in general, I would greatly appreciate two numerical experiments that simulate (a) less-quality bulletin data and (b) worse model predictions. In the first step, all data could be held equal except for the reported danger rating, which could be perturbed with a given standard deviation. In the second step, only the model predictions would be perturbed. This experiment would add another figure similar to Fig 5, which tells us (a) whether this approach will always cap model performance at the level of human performance, and (b) how a significantly worse model prediction would line up on this rather abstract visualization. This new figure could help the reader appreciate the strong results even more, and help other warning agencies to assess whether this evaluation strategy is suited for their contexts (e.g., less consistent and accurate danger rating data).

We understand this suggestion. However, this will add another layer of complexity to an already rather complex manuscript. Moreover, perturbing the human forecast isn't all that straightforward as - on a given day, the error will be the same in an aggregate of warning regions, but it may be different (or not), in other warning region aggregates. However, we will add these suggestions as possibilities in the discussion for future work. Nonetheless, rather we did think about a (somewhat simpler)

[Figure]

**Figure 1.** Model predictions. - Here an example of the natural-avalanche data displayed in Figure 4a, d, g in the manuscript, but including the bootstrap 90-% confidence interval (shading) for events and the event ratio $R$.

[Figure]

**Figure 2.** Model and bulletin predictions. - Here an example of the natural-avalanche data displayed in Figure 5a in the manuscript, but including the bootstrap 90-% confidence interval (shading).

way of exploring these issues by adding uncertainty related to the events, which are comparably rare. As we mentioned in
30 the manuscript (L342-348), the distribution of events contained the respective lowest and highest sub-levels or bins impacts the relative ratio ($RR$). We will therefore show bootstrap-sampled confidence intervals (CI) for the event data and for $RR$. Examples of the updated Figures for the natural-avalanche data are shown in Figures 1 and 2. We'll provide tables summarizing the median increase between the respective lowest and highest bins including the CI in the Appendix or as a Supplement, or appended to Table 2 in the manuscript.
35 Another comment along similar lines, but outside the scope of this manuscript unless the authors actually investigated the following thoughts already. To not run the risk of capping model performance, one could make the bins entirely independent of

the human distribution of danger ratings. The results may be less suited for comparing the model and human predictions, but we may learn better in which interval ranges the probabilities are most capable of discriminating conditions. And lastly, I fully buy into the point made by the authors that there is a limit to the value that comparing model to human data sets has, when it is unclear which data set is closer to reality when they disagree. However, I personally would be more than curious how an actual day-to-day comparison over an entire season in a prominent region looks like in the Swiss data set (for example, similar to Fig 5 in Herla et al., 2024).

With respect to the first suggestion, although potentially interesting, that's outwith the scope of our study, which focuses on a comparison. Reviewer 2 also suggested exploring a region, and as such, we will explore providing a figure for two regions covering an 8-week period during which all three models were available in forecast mode, and during which the human-made forecast sub-level varied between 1 (low) and 4+ (Figure 3). Nonetheless, an in-depth analysis is beyond the scope of this manuscript. Moreover, as event data is rare, such a region-specific analysis will at most provide an indication of trends rather than statistically robust findings. We'll use this figure to briefly highlight differences between human forecasts and model predictions, to address the challenges with verifying specific events, and to discuss the disadvantages of making a comparison using a regional context.

The storyline is sound, focused on the research objectives, and communicated well. Congratulations to the authors for this contribution!

Thank you.

**2 Detailed comments**

**2.1 Abstract**

- L14: "We conclude that these model chains are ready for systematic integration in the forecasting process." Consider adding a statement like "in Switzerland" or giving other warning agencies in other snow climates a heads-up that other modeling pipelines might not be on par.

  We will add "in Switzerland".

**2.2 Introduction**

- L47: Consider using a clearer wording, for example, *...when they independently forecast avalanche danger with a similar skill as expert forecasters*.

  We will rephrase as suggested.

- L53: I find the statement of the first objective, "(1) Is the expected increase in the number of natural avalanches or in locations susceptible to human-triggering of avalanches predicted by spatially interpolated model predictions?", more complicated than it needs to be. Consider rephrasing it, e.g., "Can spatially interpolated models predict the observed

[Figure]

**Figure 3.** Time series showing 8 weeks between 8 Feb and 7 Apr 2024 for two regions in the Swiss Alps. For the randomly selected grid points in these two regions, the respective 90%-quantile of model predictions (coloured, dashed lines and points, right y-axis) and the avalanche forecast (solid, black line, left y-axis) are shown. The shaded areas expand between the respective median value and the 90%-quantile. The bars display the number of avalanches (dark blue: natural avalanches, light blue: human-triggered avalanches).

increase in the number of natural avalanches or ...".

We will rephrase as suggested.

**2.3 Models**

70   – L80: Please add that the instability model is suited for dry slab avalanches only.

We will add this information.

– L86: Please add that the natural-avalanche model is suited for dry slab avalanches only.

We will add this information.

– L102: Downscaling weather model output to point scales is a complex endeavour. Can you please describe the key
75     modifications to the raw NWP output before you refer the reader to (Mott et al., 2023)?

The downscaling of COSMO1-input data is indeed complex and a detailed description of this process is out-of-the scope
of our manuscript as different methods are used to downscale wind, radiation, air temperature and relative humidity,
but also snow and precipitation. However, to address this we will add a statement that all of the SNOWPACK input
parameters are downscaled using a variety of approaches according to Mott et al. (2023) (Table 1).

80 ## 2.4 Data

– L111: "We analysed ..." (past tense)

We will change as suggested.

– Paragraph 3.3: In principle, the analysis would be complete with comparisons of natural and human-triggered avalanches
to a reference distribution. By including Non-events (approximated by GPS tracks), you offer another perspective on
85     evaluating performance for human triggering, a bonus so to say. Given that readers likely have a strong opinion about
using GPS tracks to approximate Non-events (see next comment), I suggest you make this point ("it's a bonus") more
clear to the reader. For me, it was helpful to understand that in Figure 2 the box "Events/Human-triggered avalanches"
caused two arrows, one to the data set that links human-triggered avalanches to the reference distribution and one to
the data set that links human-triggered avalanches to the GPS tracks. This nicely visualizes that you examined human-
90     triggering of avalanches from two complementary perspectives: one more theoretical, and the other purely data-driven,
though relying on assumptions that are not easily quantified.

– L142: GPS tracks as non-events: This approach assumes that an avalanche would have occurred if a skier loaded the
snowpack and it was unstable. That assumption holds more true for surface problems than for persistent problems buried
more deeply. In the latter case we know from avalanche accidents that it's not always the first skier who triggers the
95     avalanche, particularly since the characteristics of the slab and depth of the weak layer vary within a slope. Moreover, I
assume that the snowpack at popular ski tours or just outside of ski resorts is heavily modified by skier traffic throughout
the season. Within the typical skier corridors, weak layers will likely be destroyed and the primary avalanche problems

will likely be new snow and wind slab problems. Can you please discuss these thoughts and their potential effect on the results in the Discussion and refer to that discussion from Sect. 3.3? It would nicely add to the paragraph in L384ff.

100    There are considerable uncertainties related to the event and non-event data. This is not specific to the GPS tracks. GPS data does not contain any information about whether a slope was skied for the first time or already skied before. We only assume that no avalanche was triggered during the recorded ascent, regardless of the prevailing avalanche problem, since we think it unlikely that skiers involved in avalanches then uploaded GPS tracks. If a skier loaded the snowpack at a specific point, we can be reasonably confident - but obviously not certain - that the snowpack was sufficiently stable at
105    this location. - We'll make a remark in the respective paragraph.Off-piste terrain was excluded by only using ski touring data and also discarding data near ski lifts. We don't have robust data allowing us to systematically distinguish between popular tours and rarely used tours and will include this point in the discussion.

–    L149: Consider adding "... due to forecast, encountered avalanche conditions, or previous terrain use" (or similar).
     We will change as suggested.

110  –    L175: I suggest changing "avoid" to "minimize".
     We will change as suggested.

**2.5   Methods**

–    L195: "the random subset of grid points used as reference distribution". This is the first location that mentions that the reference distribution is based on a randomly selected subset of grid points. Please add a statement that tells the reader
115    that this concept will be explained in detail below in Sect. 4.3.
     We'll add this information as suggested.

–    Footnote 1: I think you can simply omit the footnote, particularly since you cite the same publication at the end of the sentence anyways.
     We will delete the footnote.

120  –    L203: Consider rephrasing the sentence to e.g., "For locations and elevations with dis-continuous or non-existent snow cover, we set Pr = 0."
     We will change as proposed.

–    L207: Thanks for providing the code to this analysis. Could you still summarize the high-level tuning (i.e., the hyperparameter settings) in the text of the manuscript please?
125    We are not sure which hyper-parameters these are. But we will add some more details on the settings used for the kriging method.

–    Footnote 2: Please mention the software package used to to implement the regression kriging.
     We will add the libraries used for kriging interpolation.

– L221: How sensitive are the results to the choice of 2.5% of all grid points, and how did you decide on that number? I assume the analysis is computationally fairly inexpensive. In that case, could you easily re-run the analysis for other values and report on the main differences? Also, I think the elevation filter should ideally be applied before the random sampling.

We applied the elevation filter first. To decide on the number of grid points, we looked at several subsets. In the end, we decided on 5% (2.5% is an error in the manuscript) of the 13323 grid points in the elevation range. See also Figure 4. In all four variants, the median elevation was about 2200 m. Despite the 5% variant not covering 10% of the warning regions in the Alps, the resulting proportion of sub-levels was essentially identical as for larger subsets. We therefore decided on 5% of grid points to keep computational costs low and to easily rerun experiments if necessary (interpolation for all model variants took about 5 hours on a normal laptop). - We will state clearly that we first selected grid points within the elevation range before randomly sampling. We will briefly comment on the insignificant changes in elevation and sub-level distributions between smaller and larger subsets, and that the 5% variant covers 90% of the warning regions with a median number of points per region of 5.

– L234: Consider rephrasing to e.g., "systematic biases exist between the *forecast* and *nowcast* predictions"
We will change as suggested.

– L236: "whether the models reflect the expected increase in avalanche occurrence probability with increasing model-predicted probability." I found this sentence somewhat confusing and suggest to either delete 'with increasing...' or to rewrite that last part like 'by predicting higher probabilities themselves'.
We will change as suggested.

– L241 & L243: Instead of "for cases when we relied on the reference distribution:" and "when using non-events:", please call it the same way as in Fig. 2 and 4, i.e., 'for natural and human-triggered avalanches' and 'for backcountry data'. The equations tell the reader already when the reference distributions and non-events are used.
We will change as suggested.

– L237f: Can you add a brief statement why you chose different bin widths? L254: "To obtain bins containing an equal number of data points for human forecasts and for model predictions, ...". I am not sure whether this is the correct justification. I do buy into that binning approach in order to compare human and model data, but I assume this is rather necessary because the danger rating reflects a non-linear increase in hazard (Schweizer et al., 2020; Techel et al., 2022), whereas the model predictions reflect non-linear increases of other functional shapes Mayer et al. (maybe sigmoidal?; e.g., Figure 8 in 2023). In other words, there needs to be a mapping of some sort, which you implement through the binning. Do I understand that correctly?
You understand correctly. As it is unclear how to assign probabilities describing rather different phenomena (probabilities of potential instability, of natural avalanche occurrence, or of danger levels) to sub-levels, mapping the probabilities according to frequency distributions seemed a sensible choice. As shown in Figure 5b, the derived mapping functions

[Figure]

**Figure 4.** Number of grid points used to define the representative data set. - We will not include this figure in the manuscript.

differ in shape between the models and data set. - When revising, we will include this new figure and add an explanation along these lines.

- L257: Please add ", etc." after 17.8%. That is, only if I interpret the statement "For higher sub-levels, we proceeded in the same way" correctly.

We will change as suggested.

- L266 & L268: Same comment as for L241 & L243 above.

We will change as suggested.

**2.6 Results**

- Figure 4: "... middle row: events with (d) natural avalanches, (e) human-triggered avalanches and (f) human-triggered avalanches during backcountry touring;". I don't fully understand the difference between the data used for panel (e) and (f). Can you please make that more clear.

We will make that more clear.

- Sections 5.1: Very explainable and encouraging results! Great to see it all come together after an intense workout of data acrobatics beforehand ;-)

Thank you.

- Figures 5, B1–3: I am confused why the human forecast is further stratified into the models. More specifically, why are there three lines in Fig. B1 b, d, f that are colored according to different models? As far as I understand, each panel

corresponds to one specific data set, e.g. Fig. B1d contains all natural avalanches and there should be one curve that displays the proportion of issued danger ratings. Please make sure that a correct explanation is in the text and that the reader will find that explanation from the figure captions.

As shown in Table 1, the number of cases varies between models and between *forecast* and *nowcast*, as predictions were not always available. For instance, for event type *natural avalanches*, *forecast* predictions were available on 143 days for the *natural-avalanche model*, on 236 days for the *instability model*, and on 216 days for the *danger-level model*. As the analysis is always performed for subsets of the data when model predictions and bulletin were both available, each analysis includes a different number of cases. This hardly impacts the distributions of the reference distributions or the avalanche forecast, but causes variations in the event data, resulting also in variations in the event ratio. Therefore, curves are shown for human forecasts for each model. - We will mention this more clearly in the Results section.

– Additional table: In Figures 5 and B1–3, the x-axis allows for translating between the Bin and $D_s*$. For example, Danger level 3- corresponds to bin 5. Please add a table to the Results section or Appendix, that shows the thresholds for each of the 10 bins and for each of the 3 model types, similarly to L259.

We will take up this recommendation and provide this information when introducing the approach of defining the bins (see Figure 5b). In addition, we will also provide a figure explaining the mapping approach described in Section 4.4 (Figure 5a).

**2.7  Discussion**

– L429: I suggest changing recreationists to backcountry users.

We will change as suggested.

– Paragraph 6.6: I suggest you re-iterate somewhere in the paragraph (e.g., L461) that the conclusions are valid for danger level, probability of avalanche release, and instability, but not for avalanche problems or other specific characteristics.

We will mention this as suggested.

[Figure]

**Figure 5.** (a) Assigning model-predicted probabilities (here $Pr_\text{instab}$) to bins (light-blue labels 1 to 6) equal in size to the proportion of sub-levels $D_s^*$. Shown are the cumulative proportion of $Pr_\text{instab}$ (line). Using the cumulative proportion of sub-levels, the resulting probability threshold ($thr$) can be derived. Note that proportions and thresholds for $D_s^* \geq 4-$ and bin $\geq 7$ are not shown. See explanations in the text. (b) Probability thresholds obtained for the forecast predictions for the three models using the reference distribution (ref) or GPS tracks (gps).

**References**

Herla, F., Haegeli, P., Horton, S., and Mair, P.: A quantitative module of avalanche hazard—comparing forecaster assessments of storm and persistent slab avalanche problems with information derived from distributed snowpack simulations, EGUsphere, 2024, 1–30, https://doi.org/10.5194/egusphere-2024-871, 2024.

205  Mayer, S., Techel, F., Schweizer, J., and van Herwijnen, A.: Prediction of natural dry-snow avalanche activity using physics-based snowpack simulations, Nat. Hazards Earth Syst. Sci., 23, 3445—-3465, https://doi.org/10.5194/nhess-23-3445-2023, 2023.

Mott, R., Winstral, A., Cluzet, B., Helbig, N., Magnusson, J., Mazzotti, G., Quéno, L., Schirmer, M., Webster, C., and Jonas, T.: Operational snow-hydrological modeling for Switzerland, Frontiers in Earth Science, 11, https://doi.org/10.3389/feart.2023.1228158, 2023.

Schweizer, J., Mitterer, C., Techel, F., Stoffel, A., and Reuter, B.: On the relation between avalanche occurrence and avalanche danger level,

210  The Cryosphere, https://doi.org/10.5194/tc-2019-218, 2020.

Techel, F., Mayer, S., Pérez-Guillén, C., Schmudlach, G., and Winkler, K.: On the correlation between a sub-level qualifier refining the danger level with observations and models relating to the contributing factors of avalanche danger, pp. 1911–1930, https://doi.org/10.5194/nhess-22-1911-2022, 2022.

---

## Author Comment (AC2)

**Response to review by Christoph Mitterer**

Frank Techel, and co-authors

Dear Dr. Christoph Mitterer,

we greatly appreciate your detailed and constructive feedback to our manuscript. Please find our responses below (in blue).

**1 Summary**

The study explores the effectiveness and consistency of human-generated compared to fully automated, model-driven avalanche danger forecasts by addressing two main questions:

- Does the spatial interpolation of model predictions indicate an anticipated rise in natural avalanche occurrences or an increase in areas prone to human-triggered avalanches?

- Can fully automated, data- and model-driven avalanche forecasts deliver performance levels comparable to those created by human forecasters?

In order to answer these two questions, the authors compare data sets of natural and human-triggered avalanches and GPS tracks of backcountry activity to human-made and fully automated model-driven forecasts. The human-made forecasts rely on the daily published bulletin data of Switzerland for two consecutive winter seasons (2022/2023 and 2023/2024). The model-based forecasts are explored in different modes (forecast and nowcast) for three different models (danger-level model, instability model, natural-avalanche model).

Using event ratios as proxy, the authors examine the relative accuracies and consistency of human- and machine-made forecasts by comparing the spatial interpolation of the various forecasts to (1) a reference distribution containing events/non-events, and (2) recorded events of natural and human-triggered avalanches and non-events.

The results reveal that human-made and machine-made forecasts show similar relative predictive behaviour,i.e. that increase in all model probabilities are correlated to an increase in avalanche release probability. The authors did not investigate specific or absolute behaviour. This leads the authors to the final conclusion that it is timely to introduce model-based forecasting into the operational settings of avalanche warning services.

**2 Evaluation**

The presented manuscript has a clear story line and applies in large parts transparently and comprehensible a sound set of methods to obtain innovative results in the field of model-based forecasts assessing avalanche conditions in a regional scale. The data set is innovative, methods have been already in place by the authors with other contributions (Degraeuwe et al., 2024;

Techel et al., 2022; Winkler et al., 2021). Approach and results are scientifically relevant and represent a major impact on that specific topic for the community.

Most parts of the manuscript are concise, well-structured and nicely written. Some parts though of the Data (3.1 Model predictions) and the Methods Section (4.4 Analysis) were at least for me not easy to follow. Also within the Results Sections there are part – especially the ones pointing to Table 1 that are hard to follow and grasp. In addition, the Discussion Section is very broad and remains too vague in some parts for my taste. I would advise the authors to have the courage to draw more direct conclusions so that the manuscript can have even more impact on the community for their operational approaches and future directions.

Also Figures and Tables and especially their captions will need a bit of re-touch to make them even more clear (see comments directly in the manuscript). I am convinced that this excellent work should be published on NHESS having addressed my general and specific suggestions for improvement.

**3 General comments**

My general comments touch two different aspects of the manuscript: one is of a more technical nature, the other concerns the comprehensibility of the manuscript and thus sometimes drifts into questions of taste. In this respect, I am fully aware that parts of my second comment cannot, of course, be decided objectively.

**4 Technical comments**

– The reasoning of why the authors adjusted the danger-level model of by rather opting for $Pr(D \geq 3)$ than D (Lines 110-117) including the Figure A1 is unclear. The authors refer to an "expected danger value" which might be or not connected to Pérez-Guillén et al. (2024) or to the concept of an expected danger level presented in Maissen et al. (2024). It remains, however, unclear since no citation is given. Since this altering of the danger-level model might affect central results, the reader needs more details on why the authors decided to do so. In addition to that, I would love to see what the results would look like, if the authors do not introduce this classification criteria, but demonstrate the outcome of predicting the danger level D instead.

The main reason for using $Pr(D \geq 3)$ rather than the expected danger level ($E(D)$) was that this allowed us to use exactly the same interpolation approach and subsequent assignment of rank-ordered predictions to bins as for the other models. If we had used $E(D)$, which is bound to values between 1 and 4, compared to $Pr$, bound to the range 0 to 1, the interpolation approach would have had to be adjusted considerably. - We will provide this explanation in the revised manuscript. We will make sure that terminology is consistent with either (Pérez-Guillén et al., 2024) or (Maissen et al., 2024).

– My main comment here: The authors analysed nicely now the median relative performance; is now possible to get to the more complex cases, i.e. extremes or misses of humans. The authors did a tremendous work on creating an objective test

data set. I highly acknowledge that, so why not use that approach for tackling this further question.

In this study, we focused on purpose only on the overall performance as the study is already complex and lang. However, we fully agree that shedding light on the performance of models and humans in situations, which are either rare, or where either approach fails, is an obvious next step. See also our comment below.

– With this study we have learned that the machine relatively seen thinks almost identical as a human – which in turn is only to a certain extent impressive, since at least for the danger-level model analysis are in a somewhat closed circuit: The machine learned very well to pick up the forecasting culture of a well-trained and consistent forecasting team (Switzerland) and now mimics their behaviour well in a relative way.

In this study, we moved from (a comparably small number of) point predictions, trained to predict regional-scale patterns (danger-level model, natural-avalanche model), to interpolating predictions to arbitrary locations in potential avalanche terrain (with specific slopes, aspects, and elevations). We show that this approach works, at least when compared to human forecasts for the same locations as a benchmark. Showing that human forecasts and model predictions can be interpolated to specific locations capturing the relative increase in avalanche release probability is a key first step towards higher-spatially and temporally resolved avalanche forecasts. While we have previously shown that human forecasters can predict the increasing probability of events occurring using sub-levels, at least in a relative way (Techel et al., 2022), as a side-result, we show that there is also a correlation between human forecasts and the probability of event occurrence at very specific locations.

– The most exciting question though is: when do they differ? Therefore, I would like to see at least one or two examples where either a specific region for the entire period (e.g. warning region of Davos) or a specific period (e.g. December 2023) for the entire forecasting domain.

As we also explain below, a statistical analysis of when models and forecasters differ (and fail) is not possible within this manuscript. However, as reviewer 1 suggested such an example, we have provided a figure for two regions covering an 8-week period during which all three models were available in forecast mode, and during which the human-made forecast sub-level varied between 1 (low) and 4+ (Figure 1). We'll use this figure to briefly highlight differences between human forecasts and model predictions, to address the challenges with verifying specific events, and to discuss the disadvantages of making a comparison using a regional context.

Question of taste comments:

– Please help the reader to make the two very important Sections 3.1 Model predictions and 4.4 Analysis more comprehensible. Now, sentences are very long and full of different terms referring to different "probabilities". In addition, the explanation of using $Pr(D \geq 3)$ needs more text support.

To make it easier for the reader to see that we use three different probabilities, we'll use bullet points in L128-130. In addition, we'll provide more explanation regarding the use of $Pr(D \geq 3)$ along the lines of our reply before.

[Figure]

**(a)** Northwestern Ticino (1300 km²), North aspects
Gridpoints above tree line = 27, median elevation = 2244 m

**(b)** Davos region (314 km²), North aspects
Gridpoints above tree line = 10, median elevation = 2278 m

**Figure 1.** Time series showing 8 weeks between 8 Feb and 8 Apr 2024 for two regions in the Swiss Alps. For the randomly selected grid points in these two regions, the respective 90%-quantile of model predictions (coloured, dashed lines and points, right y-axis) and the avalanche forecast (solid, black line, left y-axis) are shown. The shaded areas expand between the respective median value and the 90%-quantile. The bars display the number of avalanches (dark blue: natural avalanches, light blue: human-triggered avalanches).

[Figure]

**Figure 2.** (a) Assigning model-predicted probabilities (here $Pr_{instab}$) to bins (light-blue labels 1 to 6) equal in size to the proportion of sub-levels $D_s^*$. Shown are the cumulative proportion of $Pr_{instab}$ (line). Using the cumulative proportion of sub-levels, the resulting probability threshold ($thr$) can be derived. Note that proportions and thresholds for $D_s^* \geq 4-$ and bin $\geq 7$ are not shown. See explanations in the text. (b) Probability thresholds obtained for the forecast predictions for the three models using the reference distribution (ref) or GPS tracks (gps).

90    – Maybe you add some graphs to explain the reversed binning approaching a bit better.

We'll add a figure (Figure 2) aimed at supporting the introduction of the binning approach described in Section 4.4. (Figure 2a), and showing the shape of the functions describing the probability thresholds between bins for mapping (Figure 2b), as proposed by reviewer 1.

   – During the entire reading it was not clear to me that were working on two different human-triggered data sets (cf. Fig.
95    4e,f).

We introduced the different data sets in Section 3, providing Figure 2 as the overview linking the data and methods. We'll make this clearer.

   – Table 1 is hard to understand. E.g. what is referring to ref, what is referring to nEv?

We will revise the caption of Table 1 to explain more clearly what refers to $ref$ and what to $nEv$.

100    – I enjoyed large parts of the Discussion but have the feeling that it is too lengthy and not to the point of results that were shown. Not sure whether Section 6.4 could be shortened and incorporated into the limitation's section representing the output.

We interpolate to specific points in the start zone of avalanches. These points are defined by coordinates with 1-meter resolution. They are located in real terrain, on specific slopes. However, and on purpose, we dedicated a specific section

(Section 6.4) on the seemingly highly-resolved predictions to clearly emphasize that the interpolation approach leads to predictions which are still regional and not slope-specific. We don't consider this a limitation as such, but it is important to clearly explain the difference between resolution and scale to avoid misinterpretation. We will shorten lines L454 to 463

– I would however, love to see more work on combining Section 6.3., 6.5 by addressing the questions I posed before: When do machine and humans think differently and could this think differently help us in improving the quality of our product.

We agree that understanding when and why models and humans differ and fail is a crucial next step. However, focusing on specific situations requires analyzing very small subsets of data, which makes it difficult to conduct robust statistical analyses. Our dataset already contains a limited number of events, and even in low-frequency bins, a few occurrences are expected. These should not necessarily be considered incorrect predictions. - We plan to add a section (likely in the Discussion) where we present time series for two regions (Figure 1), briefly comparing model predictions with human forecasts. More importantly, we will address the challenges associated with verifying predictions for specific cases. The questions you pose are natural, given our results. But this paper's focus is on the performance of models across Switzerland, and we would rather see this as a logical next step rather than overfill an already complex paper.

– While reading, the feeling arises here and there that the team of authors is trying, subtly, to polarize (e.g. choice of title). Since they have done a wonderful job either way, they have no need to do so.

We don't see where we polarize but we certainly want to make the point that the performance of model pipelines have reached a state where they are approximately equal to the predictive performance of human forecasts - at least in the setup used in Switzerland.

**5  Specific comments**

See my mark-ups and comments within the attached supplement. Thank you. We will address them when revising the manuscript.

**References**

Degraeuwe, B., Schmudlach, G., Winkler, K., and Köhler, J.: SLABS: An improved probabilistic method to assess the avalanche risk on backcountry ski tours, Cold Regions Science and Technology, 221, 104 169, https://doi.org/https://doi.org/10.1016/j.coldregions.2024.104169, 2024.

Maissen, A., Techel, F., and Volpi, M.: A three-stage model pipeline predicting regional avalanche danger in Switzerland (RAvaFcast v1.0.0): a decision-support tool for operational avalanche forecasting, EGUsphere [preprint], 2024, 1–34, https://doi.org/10.5194/egusphere-2023-2948, 2024.

Pérez-Guillén, C., Techel, F., Volpi, M., and van Herwijnen, A.: Assessing the performance and explainability of an avalanche danger forecast model, https://doi.org/10.5194/egusphere-2024-2374, 2024.

Techel, F., Mayer, S., Pérez-Guillén, C., Schmudlach, G., and Winkler, K.: On the correlation between a sub-level qualifier refining the danger level with observations and models relating to the contributing factors of avalanche danger, pp. 1911–1930, https://doi.org/10.5194/nhess-22-1911-2022, 2022.

Winkler, K., Schmudlach, G., Degraeuwe, B., and Techel, F.: On the correlation between the forecast avalanche danger and avalanche risk taken by backcountry skiers in Switzerland, Cold Regions Science and Technology, 188, 103 299, https://doi.org/10.1016/j.coldregions.2021.103299, 2021.

---

## Author Response (AR1)

**Summary of major revisions**

Frank Techel, and co-authors

We sincerely thank editor Pascal Haegeli (ph) and reviewers Florian Herla (fh) and Christoph Mitterer (cm) for their extensive, thoughtful, and constructive feedback. Their comments helped us to significantly improve the clarity and focus of our manuscript.

**Summary of Major Revisions**

Given the request by the editor (ph) to simplify and restructure the manuscript, we have made several major changes, which we summarize below. Please also refer to the marked-up manuscript (track changes PDF) for detailed edits. We had already responded to the original reviewer comments in our earlier response, and have addressed all of their points in this revision, except for the two listed at the end of this reply. We use the abbreviations *fh*, *cm*, and *ph* to indicate points raised specifically by individual reviewers.

While large parts of the manuscript have been revised considerably for clarity and structure (see track changes PDF), the overall storyline, methodology, key findings, and their interpretation have not changed, or only marginally.

- **We clarified the scope and objectives** of the study, especially emphasizing that our comparison is limited to the *danger level* component of the public avalanche forecast, and that the results apply to the specific context of Switzerland. (ph, fh) This included changing the title, abstract, and introduction (60-61).

- **We simplified the structure of the manuscript** (ph), particularly in the Methods and Results sections including the following points:

    - We removed the parallel analysis of nowcasts and forecasts (ph). We now exclusively analyze predictions in forecast mode (150-151).

    - We removed the use of GPS-derived non-events for the human-triggered avalanche analysis and retained only the analysis introducing and relying exclusively on the reference distribution approach, as suggested by reviewer ph. While we believe that GPS data provide a valuable additional perspective on user exposure and complement the reference-based approach, we followed the reviewer's recommendation in order to simplify our manuscript. We briefly return to this point in the Discussion (456-463).

    - We restructured the *Data and Methods* section (Section 3) making it clearer how data and methods are linked to the two research questions (ph). For instance, Section 3.2.4 now describes explicitly the analysis and steps needed to address research question 2 (ph). Restructuring this section included revising and simplifying Figure 2 (fh, ph, cm).

- **We improved the explanation and justification of key methodological choices**, particularly in Section 3.2 (Methods), in response to reviewer requests for increased clarity and transparency (fh, cm, ph).

  - We restructured the Methods section to establish a clearer and more logical link between the analysis steps and the two research questions (ph). For instance, the steps and additional methods needed to address research question 2 are now listed in Section 3.2.4 linking them closer with this research question. (ph)
  - We expanded our explanation for choosing $\Pr(D \geq 3)$ as the danger-level model output used in the evaluation, and contrasted this with other possible outputs (Figure A1 in Appendix) (cm, ph). We added a supporting Figure A1 in the Appendix showing model behavior and decision rationale behind the use of $\Pr(D \geq 3)$. (cm)
  - We clarified the rationale for using relative event ratios in Section 3.2.3. (ph)
  - We clarified the logic and implications of our binning strategy, including why binning was necessary to enable comparison between continuous model predictions and discrete danger levels. (fh, cm)
  - We added a new explanatory figure illustrating the binning concept and approach (Figure 5a) (fh, cm) and showing all bin thresholds per model to increase transparency in Figure 5b. (fh)

- **We enhanced the clarity of figures and tables**, especially Figures 1, 2, 6, and 7, and Table 1, by improving layout, labeling, and captions. (fh, cm, ph) For instance, we now show the median model-predicted values for reference distributions and event data directly in Figure 6a-d.

- **We shortened and refocused the Discussion**, aligning and linking it more directly with our two research questions. (ph) We now begin by interpreting the key findings in relation to the two research questions (Section 5.1), moved some explanatory content to the Methods section (L244–249, Figure 4), and reduced the discussion on the potential advantages of GPS tracks over the reference distribution, as well as the limitations of the reference approach, to the Limitations section (L481–501).

Although not requested by the reviewers but we took the opportunity to **add a third forecasting season (2024/25)** to the dataset. This addition, which we discussed with the editor (ph) prior to implementation, increased the sample size and robustness of the analysis, but did not alter the overall story or key findings, nor did it increase the complexity of the manuscript.

Two suggestions by the reviewers were not implemented in this revision:

- We did not perform a simulation experiment with degraded forecast quality, randomized danger levels, or alternative binning approaches for the model predictions, as proposed by reviewer fh. While we agree that these are valuable avenues for future work, they fall outside the scope of the present study and would require additional design considerations and extensive methodological description. Nonetheless, as a partial step in this direction, we now use bootstrap sampling to derive 90%-confidence intervals for the event ratios (261-262, 276-278).

- We did not include a case-by-case analyses or time series for specific regions or periods (as suggested by reviewer cm and as originally outlined in our responses to reviewers fh and cm), in order to maintain focus on the evaluation, to avoid

expanding the manuscript further, and to follow the recommendation by reviewer/editor ph to concentrate on the main story. We agree, however, that this is a promising extension for future work.

We believe these revisions have strengthened the manuscript considerably.

---

## Editor Decision (ED1)

NHESS-2024-158

**Forecasting avalanche danger: human-made forecasts vs. fully automated model-driven predictions**

By Frank Techel, Stephanie Mayer, Ross S. Purves, Günter Schmudlach, and Kurt Winkler

April 5, 2025

Dear Frank and co-authors:

Thank you very much for your patience since our processing of this manuscript has taken much longer than it should. I have now had the time to reread your manuscript in detail and study the reviewer comments as well as your responses to them. Based on my analysis, I recommend major revisions for your manuscript. While your study provides valuable and cutting-edge insights on the value fully automated, model-driven avalanche hazard predictions, I believe considerable changes are needed to make the manuscript clearer, more accessible, and more impactful for the NHESS readership.

In addition to the comments provided by the reviewers, I would like you to consider the following comments and suggestions.

**1) Simplification of analysis**

I encourage you to simplify the manuscript as much as possible and focus it on what is really needed to address the two main objectives of the study. Right now, it seems to me that you are presenting everything that you explored for this study, but not everything might be necessary for supporting your main conclusions. For example, it is really necessary to conduct the entire analysis on nowcasts and forecasts, which seems to make Fig. 5 much more complicated than it has to be? You establish very early on that nowcast and forecast predictions are highly correlated (this could be done in the methods section since it is not related to your primary research question). Once this is established, it does not seem necessary to continuously include both in the analysis. The trends shown in Fig. 5 are almost identical and presenting them both does not seem to add much value.

Similarly, it is unclear to me what the true value is of calculating the event ratios for human triggered avalanches in two different ways: a) with the reference distribution, and b) the sum of the observed events and non-events derived from the GPS points. Both approaches produce essentially the same results (aside a few differences which you attribute to limitations of the GPS dataset), but including them both requires considerable additional explanations, assumptions, and processing steps. To me, it seems that this additional complexity only makes the study harder to understand but does not add much additional value.

These two simplifications would have several benefits: a) require fewer datasets to be introduced, b) reduces complexity of Fig. 2, c) simplify the description of the analysis (Section 4.4), d) reduce the number of panels and data series in Fig. 4, e) simplify Table 1, and f) reduce redundancies in the results section.

**2) Description of research objectives and structure of manuscript**

The research objectives stated at the end of the introduction are "(1) Is the expected increase in the number of natural avalanches or in locations susceptible to human-triggering of avalanches predicted by spatially interpolated model predictions? and (2) Do fully data- and model-driven predictions achieve performances comparable to human-made avalanche forecasts?".

If I understand your manuscript correctly, you address Objective 1 by comparing the trends in the three model predictions (instability, danger level, and natural avalanches) to trends in event ratios of natural and human triggered avalanches. You then move on to address Objective 2 by comparing the model trends again the operational avalanche danger ratings in the avalanche forecast.

First, I find the wording of the second research objective as well as the title of the manuscript rather grandiose and too general. I think it would be better for both the title and the description of Objective 2 to be very explicit about your focus on the danger rating, which is only one aspect of human-made avalanche forecasts.

Second, as pointed out by one of the reviewers, the methods and result sections are difficult to follow. I wonder whether restructuring these two sections more clearly along the two research objectives would help make the information easier to understand and more accessible to the reader. Right now, the introduction of the different dataset and description of the analysis approach seems partially out of order and a bit detached from the research objectives. Hopefully, this would allow you to introduce the different aspects of the analysis in a more logical order and not having to refer to sections across the manuscript (e.g., L137, L209). Furthermore, I hope that my earlier suggestion can help to further simplify the text.

**2) Simplification of danger rating model output**

I understand that your simplification of the danger rating model output produces a single binary variable with probabilities from 0 to 1 similar to the output of the other models, which makes your subsequent analysis easier since you can use the same approach for all models. I have not thought this all the way through, but you could do this for multiple danger rating thresholds. I am not sure whether this would provide any additional insights, but regardless, I think the methods section would benefit from a more in-depth explanation of your choice.

**3) Generalization**

Drawing universal conclusions about the ability of avalanche forecasting model chains based on a dataset from two seasons that includes considerable assumptions and simplifications without providing any measure of uncertainty is a bit challenging. I understand that the additional analysis by Florian Herla might be out of scope for this paper, but I encourage you to either a) include something along these lines to provide the reader with some sense of uncertainty in your analyses, or b) adjust the wording in your discussion and conclusion to make it clear that the results are just a description of the available dataset and generalization should only done with caution.

**4) Discussion**

The discussion seems a bit meandering and go beyond the main objectives of the study. Overall, I would appreciate a stronger focus on the discussion of the two research questions at the beginning

of the discussion before relating the results to other studies and avalanche forecasting in general. I encourage you to ground your discussion of the state of avalanche forecasting more strongly in your results and make those links more obvious throughout the text.

While I appreciate you being up front about the limitations of the study right at the beginning of the discussion section, the section primarily repeats information that was presented in the methods section already.

I would also like to encourage you not to include new analyses in discussion section (Section 6.4). In my opinion, it would be better to discuss this earlier in your manuscript since it relates to design choices you make for your study.

**Other minor comments**

**p. 3+ - Section 2:** The subheadings in Section 2 seem unnecessary since they each just consist of a single paragraph.

**p. 4 – Fig. 1:** Is it necessary to label all AWS with their elevation? It completely clutters the map, and many labels overlap and are therefore hard to read.

**p. 5 – Fig. 2:** I really appreciate these types of flow charts that explain the design of a study. However, I find this one difficult to understand. Part of the reason might be the complexity of the study (see earlier comment) and the fact that the figure is introduced at the beginning of Section 3 and never referred again later in the text. It might be helpful to talk the reader through the figure more explicitly.

**p. 6 – Section 3.2:** Please use a consistent spelling for snowline.

**p. 6+ - Section 3.3.1:** The summary statistics presented for the reference distribution, events and non-events project a level of precision that is beyond the accuracy of the data, These number could probably be rounded to the closest number of 10 m.

**p. 8+ - Footnotes:** It seems like your footnotes could be replaced with regular citations.

**p. 10 – Calculation of relative event ratio**: Please explain why calculating the relative event ratios is beneficial for the presentation.

**p. 11 – L274**: Could you please describe how you applied the chi-squared test in more detail. It is not clear to me what this adds to your analysis.

**p. 12 – Fig. 4**: Could the panels in this figure be arranged that they are more in line with the analysis approach. I believe this is the case for the first two columns (reference distribution -> events -> event ratio), but it is different for the final column where the top chart refers to non-events whereas the other two charts relate to events (human-triggered avalanches). See my earlier comment about simplifying the analysis in general. Maybe Figures 4 & 5 could even be combined.

**p. 12 – Table 1**: It is also difficult for me to completely understand the link between Fig. 4 and the numbers presented in Table 1.

---

## Author Response (AR2)

**Summary of major revisions**

Frank Techel, and co-authors

We sincerely thank editor Pascal Haegeli (ph) for providing detailed feedback on our revised manuscript. Below, please find our point-to-point reply, highlighted in blue, with changes made in the manuscript shown in red.

– L7 – Abstract: The term "reference conditions" is not clear in the abstract. It would be best to either add an explanation of the term or avoid it and describe it in different words. - We rephrased to (L7-9): We assessed the ability of both model and human forecasts to discriminate between reference distributions of conditions – typically not associated with avalanche activity – and actual avalanche events, either naturally released or triggered by humans, by calculating event ratios as proxies for release probability.

– L118: It might be useful to explicitly list the types of ML predictions to make it clear that this refers back to the danger rating, instability, and natural-avalanche models. - We rephrased to (L120): ML models like the danger-level model, the instability model, and the natural-avalanche model provide...

– L140 – Justification of Pr(D>=3): I think that some slight tweaks in this section could strengthen your justification. How about "... described in Section 2.2.1 as it simplifies the multiple probabilities model output (one probability for each danger level) into a single and more easily interpretable probability value. ... Converting the danger rating model output into single value between 0 and 1 allows us to use a consistent analysis approach for all three ML models." - We changed to (L144-147): ...described in Section 2.2.1, as it simplifies the multiple probabilities model output (one probability for each danger level) into a single and more easily interpretable probability value. Converting the danger rating model output into single value between 0 and 1 allowed us to use a consistent analysis approach for all three ML models.

– L142: Delete "and" after moreover - Done

– L158 – Description of reference grid points: I am not sure whether the explanation provided on L158-163 is necessary since you describe this in more detail (and easier to understand) just a little bit later. - We removed these lines.

– L175, 181 and 201 – Medians and IQR: I believe that the median and IQR values do not represent the precision of the actual observations accurately. I assume that the elevation values are reported to the closest 10 or 100 meters. In my opinion, the summary stats should reflect this as well, and values rounded to the closest 10 or 100 meters seem more appropriate. - Avalanches are recorded by setting a point on a map, which represents the top of the release area of the avalanche. For this point, slope aspect – with an accuracy of $\pm 25^{\circ}$ and rounded to the eight aspects used in this study – and elevation, with an accuracy of $\pm 5$ m a.s.l., are derived from a digital elevation model. We rounded to 10 m increments.

– L212: Consistent with other mentions of the research questions in the manuscript, use 'RQ' consistently. This first sentence might not be necessary anyway since it already refers to the analysis approach. - Done

– L225 – Arbitrary locations: It is unclear to me why you only extrapolate to arbitrary locations and not to all avalanche locations and reference points. I do not think this additional sampling is explained in more detail anywhere else. I am sorry if I missed it. It might be related to the bootstrapping (L262), but I am not sure. On L253, you state that you interpolated the predictions at all reference locations. - We interpolated to reference locations AND avalanche locations. We rephrase as follows to make this clearer: We spatially interpolated point data – specifically, model predictions and snowline estimates – to the locations of observed avalanche locations and to the randomly sampled reference points.

– L225 – Explanation of RK: I suggest that the first two paragraphs are combined since they discuss the same topic. Done. I also suggest that the text starting with "Compared to the simple ordinary kriging, . . ." and going all the way to the end of the paragraph is move up right at the end of the first paragraph. This fully explains the methods before you describe its application in the current study. - The entire paragraph now reads (L225-234): We spatially interpolated point data – specifically, model predictions and snowline estimates – to the locations of observed avalanche start zones and to the randomly sampled reference points in avalanche terrain. To do so, we employed regression kriging (RK) (**?**), a geostatistical method that combines a deterministic regression model with kriging of the residuals. Compared to simple ordinary kriging, RK enables the inclusion of environmental gradients, such as the varying magnitude of change with elevation. Compared to purely deterministic interpolation, it reduces bias introduced by unmodeled spatial autocorrelation. This hybrid method therefore offers improved interpolation accuracy and physical plausibility in mountainous terrain, where elevation-dependent and location-specific patterns dominate. This approach was well suited for our application, as it captures both spatial and elevational variation in avalanche conditions. In our implementation, elevation was used as a predictor in the regression component. The remaining spatial structure – unexplained by elevation – was interpolated using kriging, allowing us to better preserve local variability.

– L236 – Aspects: I am not sure whether "compound aspects" is the right term here, because NE is not a combination of N and E, it is rather the aspect between N and E. How about "intermediate aspects"? I would also expand the sentence to make it clearer why this is a challenge: . . . recorded on intermediate aspects (e.g., NE, SW) for which we did not simulate the snowpack and avalanche hazard indicators. - Changed to: intermediate aspects

– L264: I do not quite follow your sampling strategy and the binning. This relates to my earlier comment on L225. It might be useful to explain this in more detail. - We rephrased to make the sampling clearer (L263-264): To account for sampling uncertainty – especially relevant given the relatively small number of events – we applied bootstrap sampling with replacement before calculating the event ratios, repeating the procedure 100 times.

– Fig. 6 – Caption: Change to ". . . for natural avalanches (left column) and human-triggered avalanches (right column) . . . . Top row (a, b): reference distribution; middle row (c, d): . . .". - Changed to Model predictions (Pr) for natural avalanches

(left column) and human-triggered avalanches (right column). Top row: (a, b) reference distributions; middle row: (c, d)...

– L369: The first sentence might not be necessary since you already described this in the methods section. - We removed this sentence now starting: As shown in Figure **??**e, the normalized event ratio, $RR$ ((Eq. **??**)), increased markedly...

– L384: The first two sentences might not be necessary since you already described this in the methods section. Should the additional binning of the human-triggered avalanche data set already be described in the methods section? - We deleted the first two sentences. We moved the binning methodology to the Methods section 3.2.4 (L314-316).

– L398: I do not understand where the ranges are coming from. Aren't these single summary values (maybe for natural and human triggered avalanches) and not ranges? - These are single summary values for natural avalanche and the human triggered avalanche data sets for each of the three models. The "range" only refers to the range between the three models. We rephrased to make this clearer (L391-395): The average increase between adjacent bins ($\overline{F}$) ranged from 2.20 to 2.26 for the three model-specific data sets (instability, danger level, and natural avalanche model) of the human forecasts, compared to 1.63 for the instability model, 2.07 for the natural-avalanche model, and 2.0 for the danger-level model. The total increase from the lowest to highest bin ($F_{\text{total}}$) varied between 1206 and 1274 for the three model-specific data sets of the human forecasts, and from 286 (instability model) to 1163 (danger-level model).

– L401: While the difference is statistically significant, does it have practical relevance? I do not completely understand what the sample of these comparisons are? - No, we believe it is of comparably little practical relevance, beside the fact that human forecasts – integrating a wide range of data sources – still have a slight advantage in terms of discrimination power. We rephrase to make clear what we compare (L395-396): Statistically comparing the respective distributions of the 100 bootstrap samples for each of the model-specific data sets of human forecasts and models, confirmed that these differences were significant in most cases (Wilcoxon rank-sum test, $p < 0.001$), with the exception of the natural-avalanche model ($p = 0.08$).

– L466: Sykes et al. (2025) (https://nhess.copernicus.org/articles/25/1255/2025/) is another study using GPS tracks of actual terrain choices. - We now cite Sykes et al. (2025).

– L476: The fact that you are using the danger level as a proxy for probability of avalanche release even though the definition of the danger scale also includes avalanche size should probably be mentioned earlier (i.e., in Section 3.1.4). - On L54/55 (Introduction) we introduce that danger levels are based on the principle that the likelihood, number, and size of avalanches increases with increasing danger levels, and on L219-220 in Section 3.1.4, we say that forecast sub-levels correspond to greater number of locations prone to avalanche release and a higher likelihood of larger avalanches. In addition, we added (L220-221): In this study, we use the forecast sub-level as a proxy for the probability of avalanche release.